# Uncovering the heterogeneity and temporal complexity of neurodegenerative diseases with Subtype and Stage Inference

Alexandra L Young ⓘ et al.[#]

The heterogeneity of neurodegenerative diseases is a key confound to disease understanding and treatment development, as study cohorts typically include multiple phenotypes on distinct disease trajectories. Here we introduce a machine-learning technique—Subtype and Stage Inference (SuStaIn)—able to uncover data-driven disease phenotypes with distinct temporal progression patterns, from widely available cross-sectional patient studies. Results from imaging studies in two neurodegenerative diseases reveal subgroups and their distinct trajectories of regional neurodegeneration. In genetic frontotemporal dementia, SuStaIn identifies genotypes from imaging alone, validating its ability to identify subtypes; further the technique reveals within-genotype heterogeneity. In Alzheimer's disease, SuStaIn uncovers three subtypes, uniquely characterising their temporal complexity. SuStaIn provides fine-grained patient stratification, which substantially enhances the ability to predict conversion between diagnostic categories over standard models that ignore subtype ($p = 7.18 \times 10^{-4}$) or temporal stage ($p = 3.96 \times 10^{-5}$). SuStaIn offers new promise for enabling disease subtype discovery and precision medicine.

Neurodegenerative disorders, such as frontotemporal dementia (FTD) and Alzheimer's disease (AD), are biologically heterogeneous, producing high variance in in vivo disease biomarkers, such as volumetric measurements from imaging, protein measurements from lumbar puncture or behavioural measurements from psychometrics, which reduces their utility in disease studies and management. Key contributors to this heterogeneity are that individuals belong to a range of disease subtypes (giving rise to phenotypic heterogeneity) and are at different stages of a dynamic disease process (producing temporal heterogeneity). Previous studies aiming to explain biomarker variance typically focus on a single aspect of this heterogeneity: phenotypic heterogeneity at a coarse, typically late, disease stage or temporal heterogeneity in a broad population. However, the inability to disentangle the range of subtypes from the development and progression of each over time limits the biological insight these techniques can provide, as well as their utility for patient stratification. Constructing a comprehensive picture separating phenotypic and temporal heterogeneity, i.e. identifying distinct subtypes and characterising the development and progression of each, remains a major current challenge. However, such a picture would provide insights into underlying disease mechanisms, and enable accurate fine-grained patient stratification and prognostication, facilitating precision medicine in clinical trials and healthcare.

Both FTD and AD exhibit substantial pathologic, genetic and clinical heterogeneity. In FTD a large proportion of cases (around a third) are inherited on an autosomal-dominant basis, with mutations in progranulin (GRN), microtubule-associated protein tau (MAPT) and chromosome 9 open reading frame 72 (C9orf72) being the most common causes. Of the major genetic groups, GRN mutations are associated with TDP-43 type A pathology, MAPT mutations with tau inclusions, and expansions in C9orf72 with type A or type B TDP-43 pathology[1]. AD instead has a single pathological characterisation: the presence of both amyloid plaques and neurofibrillary tangles, and the proportion of autosomal-dominant cases is much smaller, accounting for between 1 and 6% of cases[2]. The pathological heterogeneity observed in AD consists of variation in the distribution of neurofibrillary tangles, with 25% of patients having an atypical distribution of neurofibrillary tangles (described as hippocampal-sparing or limbic-predominant) on autopsy at the time of death[3]. Both FTD and AD exhibit a diverse range of clinical syndromes. FTD has both behavioural and language presentations, and in genetic FTD the clinical syndromes can further include atypical parkinsonism and amyotrophic lateral sclerosis. In AD, the major clinical syndrome is broadly divided into amnestic and rarer non-amnestic variants, with non-amnestic variants including language variant AD, logopenic progressive aphasia, visuoperceptive variant AD, posterior cortical atrophy and frontal variant AD[4].

Previous studies of neurodegenerative disease heterogeneity have focussed on either temporal heterogeneity (i.e. subjects appear different at different disease stages) or phenotypic heterogeneity (i.e. distinct groups of subjects appear different even at the same disease stage), but rarely both. We refer to these two approaches as stages-only models, which account for temporal heterogeneity but not phenotypic heterogeneity, and subtypes-only models, which account for phenotypic heterogeneity but not temporal heterogeneity. Stages-only models arise for example from regression against disease stage[5,6], and data-driven disease progression modelling[7–15]. Although such models have enabled deeper understanding of the temporal progression of a range of conditions, the inherent assumption that all individuals have a single phenotype, i.e. follow approximately the same trajectory, is a key limitation. At best, this limits the biological insight and the accuracy of stratification they can provide, but potentially could also lead to erroneous conclusions. Subtypes-only models use, for example, clustering (e.g. refs. [16–23]) to identify distinct groups, or group individuals using information independent of the model, such as genetics (e.g. ref. [24]) or post-mortem examination (e.g. refs. [25–28]) for models based on in vivo imaging. With typical subtypes-only models, the limitation is the inherent assumption that all subjects are at a common disease stage so that the cohort has no temporal heterogeneity. This requires a priori staging and selection of individuals, which is typically crude in practice leaving models that are not specific to subtype differences. Models of both disease subtype and stage heterogeneity have been constructed previously for the small proportion of neurodegenerative diseases that are inherited on an autosomal-dominant basis. For example, Rohrer et al.[29] investigate temporal heterogeneity within genetic groups by regressing imaging markers against an estimated age of onset (from family history). However, such studies lack the ability to identify within-genotype phenotypes, and the temporal resolution of the recovered genotype progression patterns is limited by inaccuracy of the a priori staging.

This paper presents Subtype and Stage Inference (SuStaIn): a computational technique that disentangles temporal and phenotypic heterogeneity to identify population subgroups with common patterns of disease progression. We demonstrate SuStaIn using structural magnetic resonance imaging (MRI) data sets from cohorts of genetic FTD and AD patients. In each case, SuStaIn provides a data-driven taxonomy (set of subtypes and stages), as well as detailed pictures of the progression of neurodegeneration within each of the data-driven subgroups. From the genetic FTD data set, SuStaIn identifies subtypes from imaging alone that map closely onto the genotypes and reconstructs patterns of neurodegeneration that reflect analysis of the individual genetic groups. This provides a validation of SuStaIn's ability to identify subgroups with distinct temporal progression patterns, as the different genotypes are known to have distinct patterns of neurodegeneration visible as brain atrophy in MRI[29]. However, SuStaIn further uncovers two distinct within-genotype phenotypes for carriers of a mutation in the C9orf72 gene, while finding the MAPT and GRN mutation groups are more homogeneous. In AD, SuStaIn identifies three distinct subtypes and reconstructs their previously unseen temporal progression. In both neurodegenerative diseases, we demonstrate strong assignment of individuals to the SuStaIn subtypes, which is in contrast to subtypes-only models in the literature (e.g. ref. [23]). Even at very early stages, at least a proportion of individuals show strong alignment with particular subtypes, which highlights the potential utility in precision medicine. In AD, we show that SuStaIn subtype and stage enhance the ability to predict conversion between diagnostic categories substantially beyond subtypes-only or stages-only models.

## Results

**Subtype and stage inference.** Figure 1 provides a conceptual overview of the SuStaIn modelling technique. SuStaIn is an unsupervised machine-learning technique that identifies population subgroups with common patterns of disease progression. SuStaIn builds on and combines ideas from clustering (e.g. refs. [16–23]) and data-driven disease progression modelling (e.g. refs. [7–10,12]). The combination uniquely enables SuStaIn to group individuals with common phenotypes across the range of disease stages. It determines the number of subtypes that the available data can support, reconstructs the trajectory of stages within each subtype, and assigns a probability of each subtype and stage to each subject. These features provide insights into the underlying disease biology and a mechanism for in vivo fine-grained stratification at early disease stages.

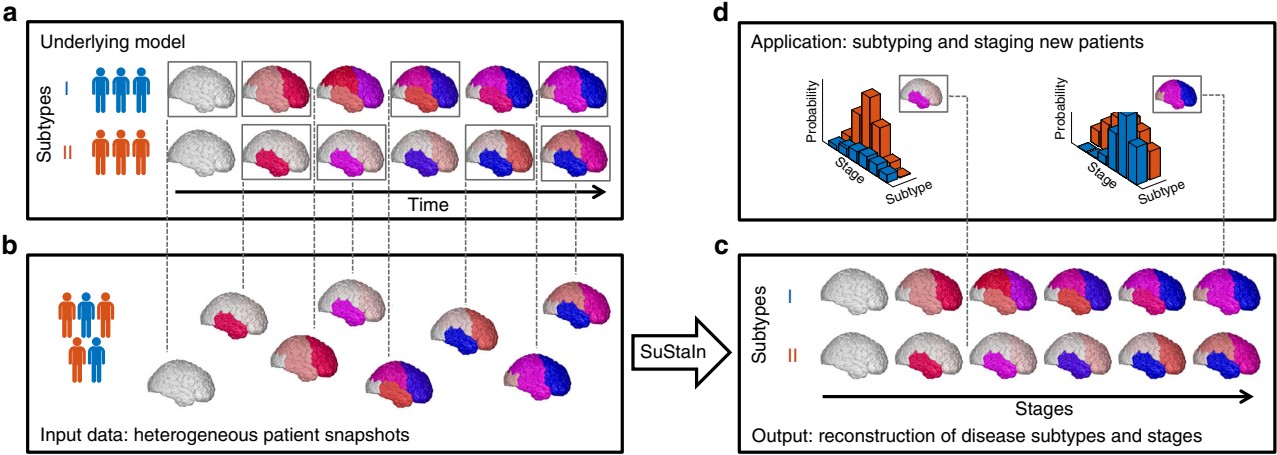

**Fig. 1** Conceptual overview of SuStaIn. The Underlying model panel (**a**) considers a patient cohort to consist of an unknown set of disease subtypes. The input data (Input data panel, **b**), which can be entirely cross-sectional, contains snapshots of biomarker measurements from each subject with unknown subtype and unknown temporal stage. SuStaIn recovers the set of disease subtypes and their temporal progression (as shown in the Output panel, **c**) via simultaneous clustering and disease progression modelling. Given a new snapshot, SuStaIn can estimate the probability the subject belongs to each subtype and stage, by comparing the snapshot with the reconstruction (as shown in the Application panel, **d**). This figure depicts two hypothetical disease subtypes, labelled I and II, and the biomarkers are regional brain volumes, but SuStaIn is readily applicable to any scalar disease biomarker and any number of subtypes. The colour of each region indicates the amount of pathology in that region, ranging from white (no pathology) to red to magenta to blue (maximum pathology)

**Synthetic data**. A simulation study (see Supplementary Methods, Supplementary Results, Supplementary Discussion and Supplementary Figures 1–12) verifies the ability of the SuStaIn algorithm to recover predefined subtypes and their progression patterns from heterogeneous data sets with comparable numbers of subjects, biomarkers and clusters (subtypes) to those used in this study.

**Subtype progression patterns**. We demonstrate SuStaIn in two neurodegenerative diseases, genetic FTD and sporadic AD, using cross-sectional regional brain volumes from MRI data in the GENetic Frontotemporal dementia Initiative (GENFI) and the Alzheimer's Disease Neuroimaging Initiative (ADNI). GENFI investigates biomarker changes in carriers of mutations in *GRN*, *MAPT* and *C9orf72* genes, which cause FTD. *GRN* and *MAPT* mutations are known to be associated with distinct phenotypes, whereas *C9orf72* is a heterogeneous group[30]. Here, GENFI serves as a test data set with a partially known ground truth for validation, as we expect SuStaIn to identify genetic groups as distinct phenotypic subtypes. However, it further supports investigation of the phenotypic and temporal heterogeneity within genotypes. Specifically, we ran SuStaIn on the combined data set from all 172 mutation carriers in GENFI (Fig. 2a), without genotypes, and compared the resulting subtype assignments and progression patterns with (a) participant's genotype labels (Fig. 2b), and (b) subtype progression patterns obtained from each genotype separately (Supplementary Figure 13; 76 *GRN* carriers, 63 *C9orf72* carriers, 33 *MAPT* carriers). Next, we used SuStaIn to identify sporadic AD subtypes from ADNI (793 subjects, including 524 with mild cognitive impairment (MCI) or AD) and characterise their progression from early to late disease stages (Fig. 3). We tested consistency of the SuStaIn subtypes in a largely independent data set—ADNI 1.5T MRI (576 subjects, including 396 with MCI or AD) scans (Fig. 4) rather than the main 3T data set used for Fig. 3. In each disease, cross-validation tests the reproducibility of the subtypes and estimated progression patterns (Supplementary Figure 14).

**SuStaIn reveals within-genotype phenotypes in FTD**. Figure 2 shows that SuStaIn successfully identifies the progression patterns of the different genetic groups in GENFI, without prior knowledge of genotype, and further suggests that phenotypic heterogeneity of the *C9orf72* group results from two neuroanatomical subtypes. Figure 2a shows the four subtypes that SuStaIn finds from the full set of all mutation carriers in GENFI. We refer to them as the asymmetric frontal lobe subtype, temporal lobe subtype, frontotemporal lobe subtype and subcortical subtype. Figure 2b reveals that *GRN* mutation carriers are the main contributors to the asymmetric frontal lobe subtype, *MAPT* mutation carriers are the main contributors to the temporal lobe subtype, and *C9orf72* mutation carriers are the main contributors to both the frontotemporal lobe subtype and the subcortical subtype. This suggests that there are two distinct subtypes in the *C9orf72* group. Application of SuStaIn to each genetic group separately supports this finding by demonstrating that the *GRN* mutation carriers are best described as a single asymmetric frontal lobe subtype, the *MAPT* mutation carriers are best described as a temporal lobe subtype and the *C9orf72* mutation carriers are best described as two distinct disease subtypes: a frontotemporal lobe subtype and a subcortical subtype. SuStaIn additionally finds a subsidiary cluster in the *MAPT* group for which the progression pattern has high uncertainty. This high uncertainty likely prevents the cluster from being detected when applying SuStaIn to all mutation carriers in Fig. 2 as this small number of subjects can be sufficiently modelled by the three alternative subtype progression patterns. Supplementary Figure 13 shows that the subtype progression patterns for each genetic group are in good agreement with those found in the full set of all mutation carriers (Fig. 2a). Supplementary Figure 14A shows that the four subtypes estimated in Fig. 2a are reproducible under cross-validation, with a high average similarity between cross-validation folds of >93% for each subtype. Altogether these results provide strong validation of SuStaIn's ability to recover distinct subtypes and their progression patterns from a heterogeneous data set, while simultaneously disentangling the heterogeneity of the *C9orf72* group into two distinct subtypes.

**SuStaIn identifies three subtype progression patterns in AD.**
Figure 3 shows the temporal progression of the three neuroanatomical subtypes that SuStaIn identifies from ADNI, which we term typical, cortical and subcortical. SuStaIn reveals that for the typical subtype, atrophy starts in the hippocampus and amygdala; for the cortical subtype in the nucleus accumbens, insula and cingulate; and for the subcortical subtype in the pallidum, putamen, nucleus accumbens and caudate. Supplementary Figure 14B shows that these three subtypes are reproducible under cross-validation, giving an average similarity between cross-validation folds of >92% for each subtype.

**AD subtypes are reproducible in an independent data set.**
Figure 4 shows that the three subtypes in Fig. 3 are reproducible in a largely independent data set (<5% subjects in common) consisting of regional brain volumes derived from 1.5T rather than 3T MRI scans. From the 1.5T data, SuStaIn broadly replicates the three major clusters found in the 3T data, again finding a typical, cortical and subcortical subtype. The origin of atrophy for each subtype is in general agreement with the 3T data: atrophy begins in the hippocampus and amygdala for the typical subtype, in the insula and cingulate for the cortical subtype; and in the pallidum, putamen and caudate for the subcortical subtype. The main difference compared to the 3T data is that the nucleus accumbens is not indicated as an early region to atrophy in the 1.5T data for the cortical and subcortical subtypes. SuStaIn

additionally identifies a small proportion (4%) of outliers with a parietal subtype in the 1.5T data.

**Disease subtyping and staging.** We investigated SuStaIn's capability for reliable stratification in each neurodegenerative disease (Fig. 5) to determine the potential for homogeneous cohort identification. First, we assessed how reliably SuStaIn assigns patients to subtypes (Fig. 5a, b). Specifically, in genetic FTD, we tested the consistency of SuStaIn subtypes with the different genotypes in symptomatic mutation carriers (Table 1), and compared this consistency against models that do not account for temporal heterogeneity (Table 2). Second, we assessed the reliability of the SuStaIn stages in each disease (Fig. 5c, d) by comparison with clinical diagnostic categories. In ADNI, where clinical follow-up information is available, we further examined the ability of SuStaIn subtypes and stages to predict relevant outcomes, by determining whether SuStaIn subtype and/or stage modify the risk of conversion between diagnostic categories (Table 3).

**SuStaIn provides utility for patient stratification.** Figure 5 illustrates the ability of SuStaIn to provide disease subtyping and staging information for each neurodegenerative disease. Figure 5a shows that the strength of assignment (see Methods: Strength of assignment to subtype) to the SuStaIn subtypes in genetic FTD increases as the diseases progress, with 88% of the symptomatic

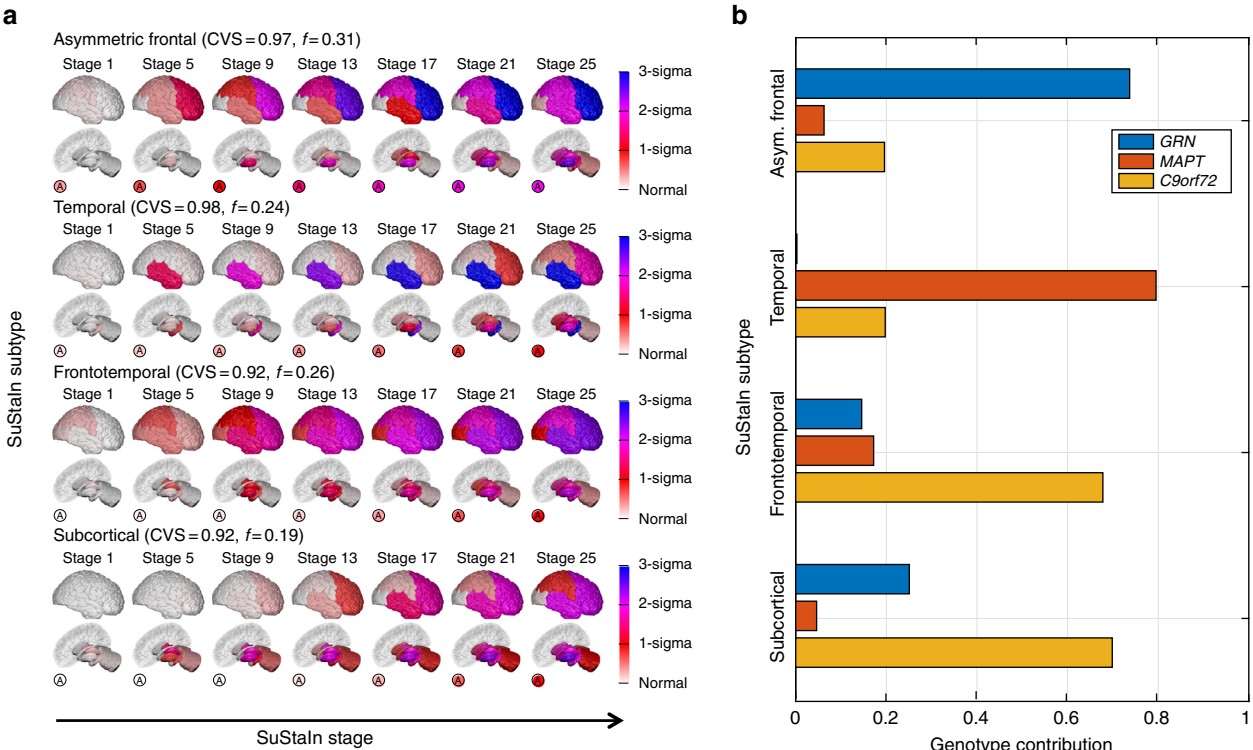

**Fig. 2** SuStaIn modelling of genetic frontotemporal dementia using GENFI data. **a** The progression pattern of each of four subtypes that SuStaIn identifies. Each progression pattern consists of a sequence of stages in which regional brain volumes in mutation carriers (symptomatic and presymptomatic) reach different z-scores relative to non-carriers. Intuitively (for a more precise description see Methods: Uncertainty Estimation), at each stage the colour in each region indicates the level of severity of volume loss: white is unaffected; red is mildly affected (z-score of 1); magenta is moderately affected (z-score of 2); and blue is severely affected (z-score of 3 or more). The circle labelled A indicates the asymmetry of the atrophy pattern (absolute value of the difference in volume between the left and right hemispheres divided by the total volume of the left and right hemispheres) at each stage for each subtype. CVS is the model cross-validation similarity (see Methods: Similarity between two progression patterns): the average similarity of the subtype progression patterns across cross-validation folds, which ranges from 0 (no similarity) to 1 (maximum similarity). f is the proportion of participants estimated to belong to each subtype. **b** The contribution of each genotype to each of the SuStaIn subtypes. This is calculated as the probability an individual has a particular genotype given that they belong to a particular subtype

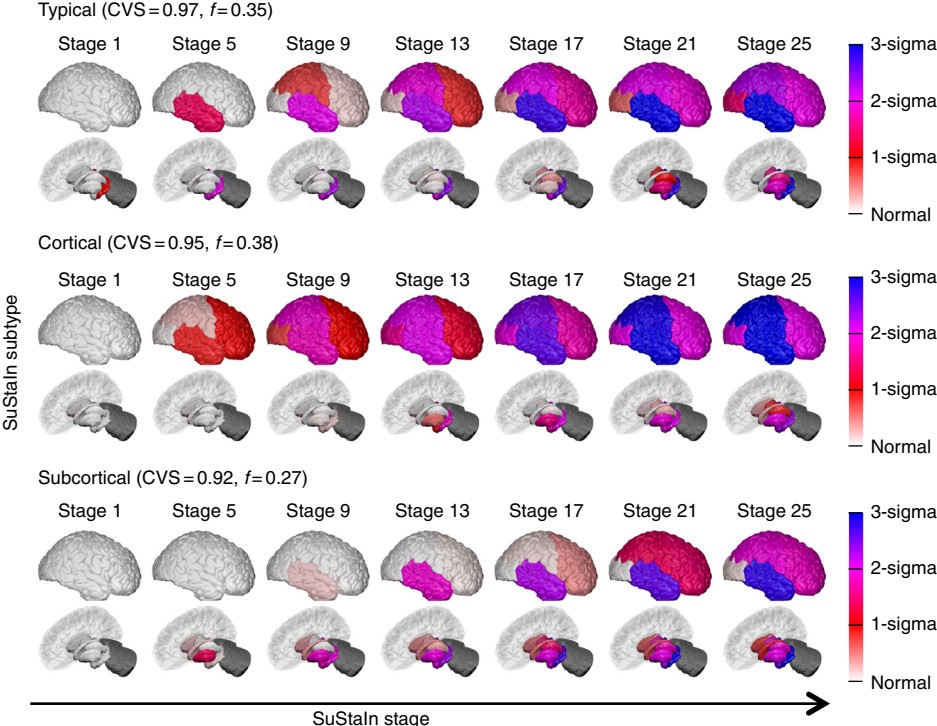

**Fig. 3** SuStaIn modelling of sporadic Alzheimer's disease using ADNI data. The rows show the progression pattern of the three subtypes identified by SuStaIn. Diagrams as in Fig. 2, but the z-scores are measured relative to amyloid-negative (cerebrospinal fluid Aβ1–42 > 192 pg per ml) cognitively normal subjects, i.e. cognitively normal subjects with no evidence of amyloid pathology on cerebrospinal fluid. The cerebellum was not included as a region in the Alzheimer's disease analysis and so is shaded in dark grey

mutation carriers in GENFI being strongly assigned (i.e. >50% likelihood of a particular subtype). Figure 5b shows that the strength of assignment to the SuStaIn subtypes in AD also increases with disease progression, with a strong assignment of individuals to subtypes in 78% of ADNI participants with an AD diagnosis. The strong assignment of the AD subtypes that SuStaIn achieves by accounting for temporal heterogeneity is in contrast to previous studies[23] that model phenotypic but not temporal heterogeneity. Moreover, the strong assignment is seen even at early disease stages (MCI), where many subjects cluster around the vertices of the triangles: 37% of MCI subjects are strongly assigned to a subtype. Figures 5c, d show that the distribution of SuStaIn stages differs between diagnostic groups in both GENFI and ADNI, and provides a good separation of presymptomatic and symptomatic mutation carriers, and cognitively normal (CN) and AD.

**SuStaIn subtypes discriminate FTD genotype**. Table 1 shows the classification accuracy obtained using the SuStaIn subtypes in Fig. 2 to discriminate the genotype of affected mutation carriers in GENFI. While the use of MRI to identify genotype is not necessary in these subjects, so not clinically relevant, this experiment demonstrates the ability of SuStaIn to identify subtypes in a data set with a known ground truth. The SuStaIn subtypes give a balanced accuracy of 95% for the two-way classification task of distinguishing the homogeneous GRN and MAPT carrier groups. For the more challenging three-way classification task of distinguishing all genotypes in the presence of heterogeneity, the SuStaIn subtypes provide a maximum balanced accuracy of 86%. A high proportion of the homogeneous GRN and MAPT carrier groups are correctly assigned to the asymmetric frontal lobe (93% of affected GRN carriers) and temporal lobe subtype progression patterns (91% of affected MAPT carriers). The heterogeneous C9orf72 carrier group are much more

difficult to classify, with a total of 75% of affected C9orf72 carriers being assigned to the frontotemporal lobe and subcortical subtypes. Apart from heterogeneity, the C9orf72 carriers are also more difficult to classify because the frontotemporal lobe and subcortical subtype progression patterns are more similar to the other subtypes; by evaluating the similarity of each pair of subtype progression patterns (see Methods: Similarity between two subtype progression patterns) we find that the asymmetric frontal lobe and temporal lobe subtypes have the most distinct progression patterns of any pair of subtypes; the asymmetric frontal lobe and frontotemporal lobe subtypes have the most similar progression patterns of any pair of subtypes. The precise strategy of assigning subjects to subtype can alter the classification rates somewhat and Supplementary Table 1 examines this effect.

**Genotype discrimination out-performs subtypes-only models**. Table 2 shows the classification accuracy obtained using a subtypes-only model (Fig. 6), which does not account for temporal heterogeneity, to discriminate the genotype of affected mutation carriers in GENFI. The SuStaIn subtypes out-perform the subtypes-only model. The subtypes-only model gives a balanced accuracy of 92% compared to 95% using SuStaIn for the two-way classification task of distinguishing GRN and MAPT carrier groups; the subtypes-only model gives a maximum balanced accuracy of 69% compared to 86% using SuStaIn for the three-way classification task of distinguishing all genotypes. In the subtypes-only model the majority of misclassifications arise from the earlier stage affected GRN and MAPT carriers being assigned to the mild frontotemporal subtype associated with C9orf72 carriers. See also Supplementary Table 2.

**SuStaIn subtypes and stages have predictive utility in AD**. Table 3 shows that the SuStaIn subtypes and stages have

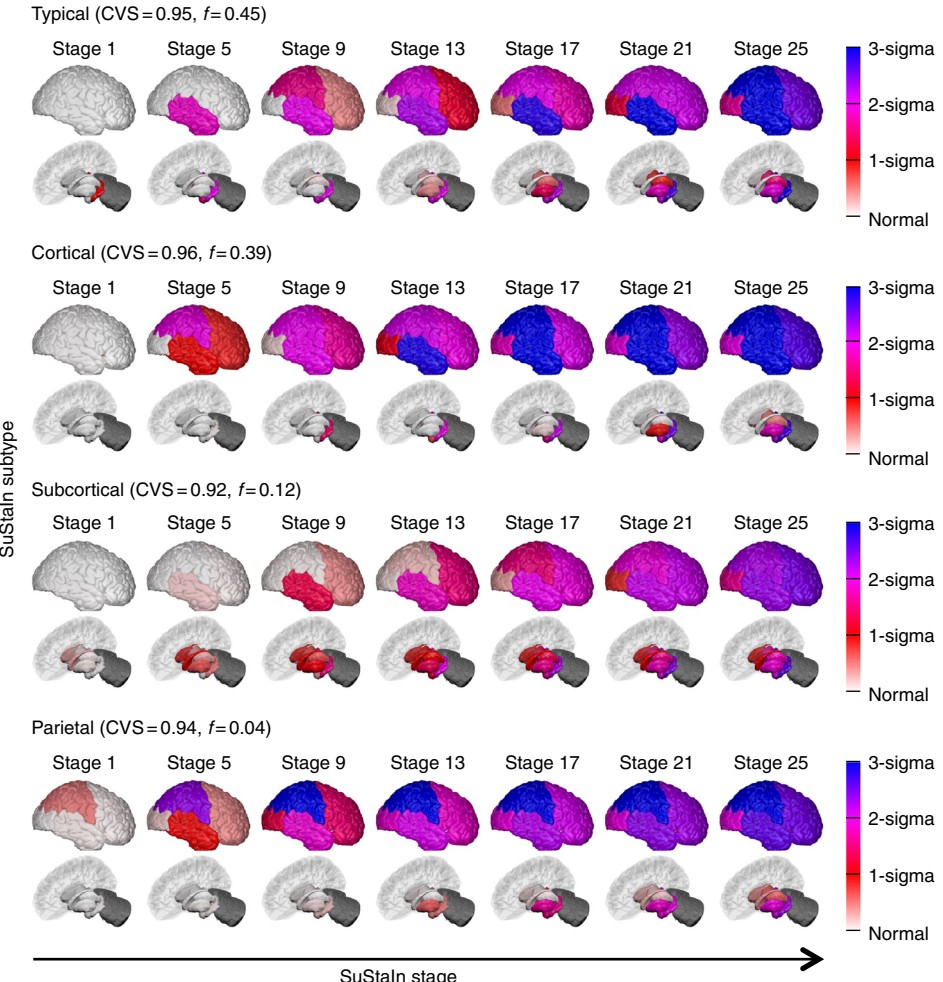

**Fig. 4** Reproducibility of the SuStaIn subtypes in Fig. 3. A largely independent Alzheimer's disease data set (only 59 subjects are in both the 576 subject data set used to generate this figure and the 793 subject data set used in Fig. 3) consisting of those with regional brain volume measurements from 1.5T MRI scans, rather than 3T MRI scans, are shown. Diagrams are as in Fig. 3, with the rows showing the progression pattern of one of the subtypes identified by SuStaIn. SuStaIn modelling identifies three major subtypes: a typical, a cortical and a subcortical subtype, which are in good agreement with the three subtypes in Fig. 3, as well as an additional very small outlier group (only 4%) with a subtype we term parietal. This small subgroup may represent outliers with a posterior cortical atrophy phenotype

predictive utility for the risk of conversion between diagnostic categories in ADNI. By fitting a Cox Proportional Hazards model, we found significant effects (*t*-test) of baseline SuStaIn subtype ($p = 2.44 \times 10^{-3}$) and stage ($p = 8.76 \times 10^{-11}$) on an individual's risk of conversion from MCI to AD. Of the SuStaIn subtypes, the subcortical subtype is associated with the lowest risk of conversion, while the typical subtype is associated with the highest risk of conversion. Supplementary Table 3 shows that SuStaIn outperforms subtypes-only and stages-only models at estimating the risk of conversion between diagnostic categories in ADNI. By performing likelihood ratio tests comparing SuStaIn to subtypes-only and stages-only we find that SuStaIn provides a significantly better fit (likelihood ratio test) than both subtypes-only ($p = 3.96 \times 10^{-5}$) and stages-only ($p = 7.18 \times 10^{-4}$) models. This shows that both the subtypes and stages estimated by SuStaIn provide additional information for predicting the risk of conversion from MCI to AD.

## Discussion
In this study we introduce SuStaIn—a powerful tool for data-driven disease phenotype discovery, providing insights into disease aetiology, and enhanced power for patient stratification in clinical

trials and healthcare. Results from the GENFI data set first validate that SuStaIn can successfully recover known distinct progression patterns in genetic FTD corresponding to different genotypes. Moreover, SuStaIn identifies and characterises within-group heterogeneity for carriers of a mutation in the *C9orf72* gene as distinct temporal progression patterns in two subtypes. The results demonstrate the utility of SuStaIn for data-driven disease phenotype discovery, and provide biological insight into the *C9orf72* mutation. Application of SuStaIn to the 3T ADNI data set recovers three distinct AD subtypes with final stages that reflect post-mortem neuropathological findings. Results from a largely independent AD data set (ADNI 1.5T) corroborate these three subtypes. The disease subtype characterisation SuStaIn provides goes much further than post-mortem neuropathological studies[3,28], or other machine-learning techniques[23], by characterising the temporal trajectory of each subtype, enabling in vivo stratification of subjects by disease stage as well as disease subtype. We demonstrate the ability of SuStaIn to stratify in vivo by both subtype and stage in genetic FTD and AD. In genetic FTD, we show that the SuStaIn neuroimaging subtypes can distinguish affected carriers belonging to different genetic groups with high classification accuracy. In AD, we demonstrate strong assignment of subjects to the SuStaIn subtypes

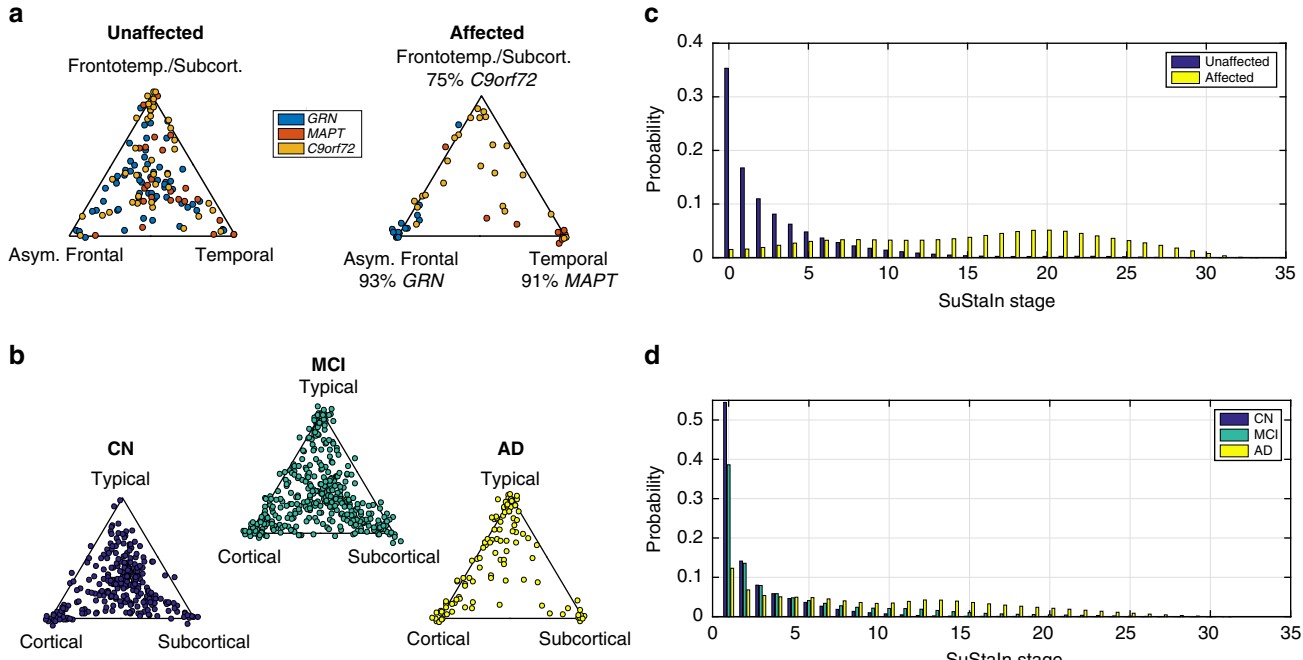

**Fig. 5** SuStaIn subtyping and staging of genetic frontotemporal dementia and Alzheimer's disease. **a**, **b** The assignability of the disease subtypes estimated by SuStaIn for genetic frontotemporal dementia, and Alzheimer's disease. Each scatter plot visualises the probability that each individual belongs to each of the SuStaIn subtypes estimated for **a** genetic frontotemporal dementia (as shown in Fig. 2a), and **b**. Alzheimer's disease (as shown in Fig. 3). In the triangle scatter plots, each of the corners corresponds to a probability of 1 of belonging to that subtype, and 0 for the other subtypes; the centre point of the triangle corresponds to a probability of 1/3 of belonging to each subtype. **c**, **d** The probability subjects from each of the diagnostic groups belong to each of the SuStaIn stages for **c** genetic frontotemporal dementia and **d** Alzheimer's disease. CN = cognitively normal; MCI = mild cognitive impairment; AD = Alzheimer's disease

| Table 1 Ability of subtypes to distinguish between different genotypes in symptomatic mutation carriers in GENFI using the SuStaIn subtypes in Fig. 2a | | | |
|---|---|---|---|
| | **GRN** | **MAPT** | **C9orf72** |
| Asymmetric frontal (threshold $p > 0.65$) | 93% (13) | 9% (1) | 4% (1) |
| Temporal (threshold $p > 0.35$) | 0% (0) | 91% (10) | 21% (5) |
| Frontotemporal | 0% (0) | 0% (0) | 42% (10) |
| Subcortical | 7% (1) | 0% (0) | 33% (8) |
| Accuracy | 93% (13/14) | 91% (10/11) | 75% (18/24) |

Each entry is the percentage (number) of participants of a particular genotype assigned to that subtype. The final row indicates the percentage (fraction) of participants assigned to the correct subtype from each genotype. The results show that SuStaIn can accurately discriminate genotypes, validating the ability of SuStaIn to identify distinct phenotypes that align with known genetic groups

| Table 2 As Table 1, but for subtypes obtained from a subtypes-only model that accounts for heterogeneity in disease subtype but not disease stage, i.e. the subtypes in Fig. 6 | | | |
|---|---|---|---|
| | **GRN** | **MAPT** | **C9orf72** |
| Severe frontal (threshold $p > 0.99$) | 57% (8) | 9% (1) | 4% (1) |
| Severe temporal (threshold $p > 0.99$) | 0% (0) | 64% (7) | 8% (2) |
| Mild frontotemporal | 43% (6) | 27% (3) | 88% (21) |
| Accuracy | 57% (8/14) | 64% (7/11) | 88% (21/24) |

The results show that SuStaIn (Table 1) provides much better discrimination of the different genotypes than the subtypes-only model shown here, demonstrating the added utility of a model that accounts for heterogeneity in disease stage

even at early disease stages (MCI), and that the SuStaIn subtypes and stages have added utility for predicting conversion between clinical diagnoses, beyond more traditional stages-only or subtypes-only models.

Previous studies in genetic FTD have found asymmetric frontotemporoparietal lobe volume loss in *GRN* carriers, temporal lobe volume loss in *MAPT* carriers, and widespread symmetric grey matter atrophy and volume loss in the cerebellum in *C9orf72* carriers[24]. The asymmetric frontal lobe subtype and temporal lobe subtype in Fig. 2 show clear similarities with previous studies of regional volume loss in *GRN* and *MAPT* mutation carriers respectively. However, SuStaIn provides much greater detail and accuracy by avoiding reliance on crude a priori

staging, e.g. via mean familial age of onset. The frontotemporal lobe subtype and subcortical subtype in Fig. 2 both have features previously associated with *C9orf72* mutation carriers, but SuStaIn assigns these features to two distinct disease subtypes, and further reveals the temporal progression of each subtype.

Several biological factors may produce the two subtypes observed in *C9orf72* mutation carriers, either individually or in combination. Clinically, while there is significant overlap, patients typically present with either a behavioural variant FTD or amyotrophic lateral sclerosis as their main phenotype[31], and they can progress at various rates; genetically, the expansion length is variable and there are additional genetic modifiers (e.g. *TMEM106B* and *ATXN2*) that alter phenotype[32–34]; and pathologically, most cases have either type A or type B TDP-43

**Table 3 Utility of SuStaIn subtype and stage for predicting the risk of conversion from mild cognitive impairment to Alzheimer's disease**

|  | SuStaIn subtype | SuStaIn stage | Age | Sex | Education | APOE4 |
|---|---|---|---|---|---|---|
| S–C–T | 1.57** | 1.13† | 0.98 | 0.98 | 0.93⁻ | 1.82† |
| S–C | 1.76⁻ | 1.16† | 0.95* | 1.03 | 0.92 | 1.53* |
| C–T | 1.48⁻ | 1.11† | 0.98 | 0.87 | 0.97 | 1.84† |
| S–T | 2.11* | 1.13† | 1.02 | 1.13 | 0.90* | 2.13† |

Each row shows Hazards ratios for a different Cox Proportional Hazards model that estimates the risk of conversion from mild cognitive impairment to Alzheimer's disease using ADNI data. Each column shows the estimated hazard ratio for each variable. Each hazards ratio tells you how the risk of conversion changes for each unit increase of a particular variable: a ratio of 1 means no modification of the risk, a ratio >1 means there is an increase of the risk, and a ratio less than 1 means there is a reduction of the risk. For the first model (S–C–T) it is assumed that the hazard ratio increases multiplicatively from the Subcortical subtype (S) to the Cortical subtype (C) to the Typical subtype (T), i.e. the S–C–T model estimates that each SuStaIn subtype has a hazards ratio 1.57 times that of the previous subtype (i.e. the cortical group have a 1.57 times greater risk of conversion than the subcortical group, the typical group have a 1.57 times greater risk of conversion than the cortical group, and the typical group have a 2.46 (1.57²) times greater risk of conversion than the subcortical group). In the remaining models only two groups are compared at a time to demonstrate that the results are similar without this assumption, although the statistical power is reduced. This result demonstrates the added utility of both disease subtypes and stages obtained from SuStaIn for predicting conversion between mild cognitive impairment and Alzheimer's disease, with both subtype and stage modifying the risk of conversion.
Statistical significance is indicated as: ⁻$p < 0.1$, *$p < 0.05$, **$p < 0.01$, †$p < 1 \times 10^{-3}$

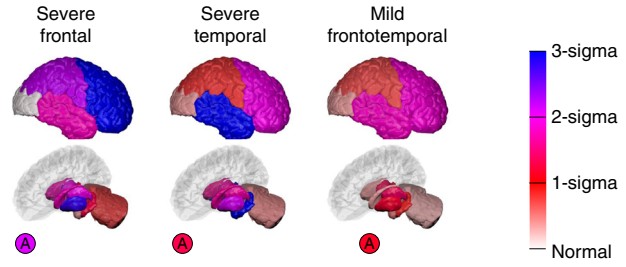

Severe frontal    Severe temporal    Mild frontotemporal

3-sigma
2-sigma
1-sigma
Normal

**Fig. 6** Subtypes-only model for GENFI; not accounting for disease stage heterogeneity. Brain diagrams as in Fig. 2, but here each diagram represents a different subtype, which we refer to as severe frontal, severe temporal and mild frontotemporal. There is no notion of disease stage in the subtypes-only model

pathology[31]. While further study is required to determine the biological factors that influence neuroanatomical phenotype, these findings demonstrate the power of SuStaIn in identifying hitherto unrecognised disease subtypes using clinical data, and thus open up the potential to link variations in genetics, pathology and neuroanatomy.

We also find evidence for the presence of a subsidiary group in the *MAPT* mutation carriers, but numbers are too small to determine whether this group have a distinct progression pattern. Among individuals with significant evidence of MRI atrophy (SuStaIn stage of ≥5), four individuals (two pairs of individuals from the same families) of 13 were identified as belonging to the subsidiary group. Although *MAPT* mutations have been commonly thought to have a very specific pattern of atrophy affecting the anterior and medial temporal lobes predominantly, one previous paper has shown that there can be a second pattern of atrophy in specific mutations, where the lateral temporal lobes are affected more than the medial regions[35]. Interestingly, the two pairs of individuals who constitute the subsidiary group in our analysis all have *P301L* mutations, a mutation that falls into this second alternate atrophy pattern group in ref. [35]. None of the nine individuals assigned to the predominant progression pattern in our analysis have *P301L* mutations, or *V337M* mutations, the other mutation identified in ref. [35] as having an alternate atrophy

pattern. This suggests that SuStaIn may be able to identify particular *MAPT* mutations that fall into this alternate group, but larger studies will be required to confirm this.

In AD, post-mortem histology[3] and retrospectively analysed MRI scans close to the time of death[28] observe three distinct patterns of atrophy in late-stage AD patients: one focussed on the temporal lobe that is similar to the late stages of the typical SuStaIn subtype; one affecting predominantly cortical regions cf. late stages of the cortical SuStaIn subtype; and one with stronger subcortical involvement cf. late stages of the subcortical SuStaIn subtype. This gives confidence in the SuStaIn subtypes, which provide much greater information by revealing the progression of each subtype over time, including the earliest sites of regional volume loss. Moreover, and importantly for practical utility, the SuStaIn subtypes can be assigned in vivo using MRI, enabling linkage of late-stage pathological observations with early-stage neurodegeneration.

The three AD subtypes found in the 3T MRI data set are corroborated by the largely independent 1.5T MRI data set. However, some small differences arise between the subtype progression patterns of the three subtypes recovered in each data set. These differences lie predominantly in how early the nucleus accumbens begins to atrophy in the different subtypes: across all three subtypes the nucleus accumbens shows atrophy earlier in the 3T subtypes than the 1.5T subtypes. A possible explanation for this is that the volume of the nucleus accumbens, which is relatively small, can be estimated more accurately using the higher field strength 3T MRI scans than the 1.5T MRI scans, and thus atrophy in the nucleus accumbens can be identified from an earlier stage in the 3T data set compared to the 1.5T data set.

In the 1.5T MRI data set we additionally find a small proportion (4%) of outliers with a parietal subtype. This small subgroup may represent a posterior cortical atrophy phenotype: comparing the Alzheimer's disease Assessment Scale-cognitive subscale (ADAS-cog) scores between individuals with an AD diagnosis that are assigned to the parietal subgroup ($N = 6$) and the typical AD subgroup ($N = 65$), we find that the parietal subgroup have worse performance (Mann–Whitney $U$ test) on certain praxic (Q6. Ideational Praxis, $p = 6.1 \times 10^{-3}$, $z = 2.7$) and spatially-demanding (Q14. Number Cancellation, $p = 4.9 \times 10^{-3}$, $z = 2.8$) subtests, but similar performance (Mann–Whitney $U$ test) in memory domains (Q8. Word Recognition, $p = 0.81$, $z = -0.2$; Q1. Word Recall, $p = 0.48$, $z = 0.70$). Additionally, the parietal subgroup is on average 10.3 years younger ($p = 2.8 \times 10^{-3}$, $z = -3.0$, Mann–Whitney $U$ test) than the typical AD subgroup.

The temporal spreading patterns for distinct subtypes estimated by SuStaIn offer biological insight. For example, the progression pattern of each subtype provides a view of how neurodegeneration spreads from a distinct origin over the rest of the brain that is uncorrupted by phenotypic heterogeneity. A key advantage of SuStaIn is that it provides a purely data-driven, hypothesis-free, reconstruction of the progression of neurodegenerative disease subtypes. However, these observations also have great potential to inform mechanistic models[36,37] of neurodegenerative disease, which explain their temporal progression via various hypothetical mechanisms of disease propagation over brain networks. Current mechanistic models implicitly assume a single-disease progression pattern—an assumption often violated in patient data sets, but much more reasonable if focussed on particular SuStaIn subtypes.

SuStaIn shows strong capabilities for patient stratification in AD, which we are able to validate in genetic FTD where we expect the subtype assignments to correspond to distinct genotypes. SuStaIn provides high classification accuracy for differentiating the different mutation types in genetic FTD, and the AD subtypes are clearly assignable. In genetic FTD, SuStaIn out-performs a

subtypes-only model, giving a balanced classification accuracy of 86% for distinguishing genotype compared to 69% for the subtypes-only model. This provides compelling evidence that there is substantial heterogeneity in disease stage within different phenotypes, and that modelling this disease stage heterogeneity is important for better patient stratification. This is further demonstrated in AD, in which SuStaIn's subtypes and stages substantially out-perform subtypes-only and stages-only models for predicting conversion between diagnostic categories. These early results are highly promising, particularly given that the particular choice of biomarkers used here (coarse regional brain volumes) is not optimised for stratification. In this initial study, we chose to use MRI to maximise the number of subjects with all available measurements, to simplify the interpretation of the results, and to enhance the clinical utility. However, future work will test the added benefit of including a wider range of bio-markers and a more fine-grained set of regional volumes in SuStaIn for patient stratification. For example in AD, incorporation of amyloid and neurofibrillary tangle measures, e.g. from amyloid and tau positron emission tomography (PET) scans, will enable stratification of individuals at the very earliest disease stages.

The previous study of Zhang et al.[23] also looked at the assignment of AD subtypes using a subtypes-only model that does not account for temporal heterogeneity in disease stage. In contrast to the study of Zhang et al.[23], we observe strong assignment of AD patients to the subtypes (Fig. 5b) emphasising the importance of accounting for heterogeneity in disease stage. This assignability clearly increases with disease progression, with the subtypes being most strongly assigned in clinically diagnosed AD patients. However, even at early stages (MCI), many subjects cluster around the vertices of the triangles showing strong potential for identifying early-stage cohorts representative of each subtype.

The model underlying SuStaIn makes several assumptions to enable the simultaneous estimation of subtypes and their pro-gression. One assumption is that biomarker variance is inde-pendent. In reality biomarkers tend to co-vary due to shared biological processes. However, simulation experiments (Supple-mentary Figure 7) show that the subtype progression patterns recovered by SuStaIn are robust to biomarker covariance. Nevertheless, refinements might come from modelling covariance among strongly dependent biomarkers, e.g. using model selection criteria to identify a minimal set of necessary covariance para-meters, and future work will explore this idea.

To enable the modelling of purely cross-sectional data, here we make an assumption of an arbitrary timescale. This formulation can also work with longitudinal data when available, although here we reserve the longitudinal information to validate the clinical utility of SuStaIn to make future predictions at an indi-vidual's first visit. However, extensions to SuStaIn that utilise longitudinal information to further provide a well-defined time-scale are an important area for future work.

Here we make a further implicit assumption that the cohort is correctly diagnosed. While the genetic tests in GENFI ensure this, the clinical diagnoses of AD in ADNI are less reliable and the proportion of misdiagnosis, e.g. of depression or other neurolo-gical diseases, is non-negligible. Simulation of the effect of mis-diagnosis (Supplementary Figure 9) demonstrates that SuStaIn can robustly recover subtype progression patterns under a sub-stantial proportion of outliers, up to 20%. Nevertheless, future adaptions of the SuStaIn model might include a broad outlier class to capture individuals that do not fit any of the main clusters.

One caveat on the findings is that the underlying data may come from a spectrum of disease progression patterns, rather

than a set of distinct trajectories as SuStaIn is designed to esti-mate. Simulations (Supplementary Figure 11) demonstrate that SuStaIn may still recover multiple distinct progression patterns from data generated by a spectrum of progression patterns. In this case the distinct progression patterns identified by SuStaIn still provide useful information about the extrema within the underlying spectrum of progression patterns. Here, however, the alignment with genotypes in genetic FTD and neuropathological observations in AD provide confidence that the distinct subtypes are genuine. Future work will extend SuStaIn to be able to represent spectra of progression patterns (e.g. using Mallow's models as in refs. [38,39]).

We introduce SuStaIn—a tool to disentangle and characterise the temporal and phenotypic heterogeneity of neurodegenerative diseases. We use it to elucidate the temporal and phenotypic heterogeneity of both genetic FTD and AD subtypes with pre-viously unseen detail. We further demonstrate SuStaIn's potential as a patient stratification tool in AD by showing strong alignment of subjects with specific subtypes even at early disease stages, as well as added power to predict conversion between clinical diagnoses. SuStaIn has the potential to make substantial clinical impact as a tool for precision medicine and is readily applicable to any progressive disease, including other neurodegenerative dis-eases, respiratory diseases and cancers.

## Methods

**GENFI data set.** We used cross-sectional volumetric MRI data from GENFI (http://www.genfi.org.uk/). Subjects were included from the second data freeze of GENFI, which in total consisted of 365 participants recruited across 13 centres in the United Kingdom, Canada, Italy, The Netherlands, Sweden and Portugal. A total of 313 participants had a usable volumetric T1-weighted MRI scan for analysis (15 participants did not have a scan and the other participants were excluded as the scans were of unsuitable quality due to motion, other imaging artefacts or pathology unlikely to be attributed to FTD). The 313 participants included 141 non-carriers, 123 presymptomatic carriers and 49 symptomatic carriers. Of the 123 presymptomatic mutation carriers there were 62 GRN, 39 C9orf72 and 22 MAPT carriers. Of the 49 symptomatic carriers, there were 14 GRN, 24 C9orf72 and 11 MAPT carriers. The acquisition and post-processing procedures for GENFI have been previously described in ref. [29]. Briefly, cortical and subcortical volumes were generated using a multi-atlas segmentation propagation approach[40], combining cortical regions of interest to calculate grey matter volumes of the entire cortex, separated into the frontal, temporal, parietal, occipital, cingulate and insula cor-tices. In addition to regional volumetric measures, we also included a measure of asymmetry, which is calculated as the absolute value of the difference between the volumes of the right and left hemispheres, normalised by the total volume of both hemispheres. This asymmetry measure was log transformed to improve normality. See Supplementary Table 4 for a summary of the biomarkers used in the SuStaIn modelling.

**ADNI data set.** Data used in the preparation of this article were obtained from the Alzheimer's Disease Neuroimaging Initiative (ADNI) database (http://adni.loni.usc.edu). The ADNI was launched in 2003 by the National Institute on Aging (NIA), the National Institute of Biomedical Imaging and Bioengineering (NIBIB), the Food and Drug Administration (FDA), private pharmaceutical companies and non-profit organisations, as a $60 million, 5-year public-private partnership. For up-to-date information, see http://www.adni-info.org. Written consent was obtained from all participants, and the study was approved by the Institutional Review Board at each participating institution.

We downloaded data from Laboratory Of Neuro Imaging (LONI; http://adni.loni.usc.edu) on 11 May 2016 and constructed two cross-sectional volumetric MRI data sets for SuStaIn model fitting: those with higher (3T) and lower (1.5T) field strength. The inclusion criteria for the 3T and 1.5T data sets were having cross-sectional FreeSurfer volumes available that passed overall quality control from either a 3T (processed using FreeSurfer Version 5.1) or a 1.5T (processed using FreeSurfer Version 4.3) MRI scan. The 3T data set consisted of 793 subjects (183 CN, 86 significant memory concern, 243 early MCI, 164 late MCI, 117 AD), of which 73 were enroled in ADNI-1, 99 were enroled in ADNI-GO and 621 were enroled in ADNI-2. The 1.5T data set consisted of 576 ADNI-1 subjects (180 CN, 274 late MCI, 122 AD). The 1.5T and 3T data sets are largely independent: only 59 subjects (14 CN, 33 late MCI, 12 AD) have both 1.5T and 3T scans. We downloaded processed cross-sectional FreeSurfer volumes for 1.5T and 3T scans, using FreeSurfer Versions 4.3 and 5.1, and quality control ratings. We retained only the volumes that passed overall quality control, and normalised them by regressing against total intracranial volume. We further downloaded demographic

information for covariate correction: age, sex, education and *APOE* genotype from the ADNIMERGE table. We downloaded diagnostic follow-up information to test the association of the SuStaIn model subtypes and stages with longitudinal outcomes. We also downloaded baseline cerebrospinal fluid (CSF) measurements of Aβ1–42, which we used to identify a control population. Again, see Supplementary Table 4 for a summary of the biomarkers used in the SuStaIn modelling.

**z-scores**. We expressed each regional volume measurement as a *z*-score relative to a control population: in GENFI we used data from all non-carriers, in ADNI we used amyloid-negative CN subjects, defined as those with a CSF Aβ1–42 measurement >192 pg per ml[41]. This gave us a control population of 48 amyloid-negative CN subjects for the 3T data set, and 56 amyloid-negative CN subjects for the 1.5T data set. We used these control populations to determine whether the effects of age, sex, education or number of *APOE4* alleles (ADNI only) were significant, and if so to regress them out. We then normalised each data set relative to its control population, so that the control population had a mean of 0 and standard deviation of 1. Because regional brain volumes decrease over time the *z*-scores become negative with disease progression, so for simplicity we took the negative value of the *z*-scores so that the *z*-scores would increase as the brain volumes became more abnormal.

**SuStaIn modelling**. We formulate the model underlying SuStaIn as groups of subjects with distinct patterns of biomarker evolution (see Mathematical Model). We refer to a group of subjects with a particular biomarker progression pattern as a subtype. The biomarker evolution of each subtype is described as a linear *z*-score model in which each biomarker follows a piecewise linear trajectory over a common timeframe. The noise level for each biomarker is assumed constant over the timeframe and is derived from a control population (see Mathematical model). This linear *z*-score model is based on the event-based model in refs. [7,8,38], but reformulates the events so that they represent the continuous linear accumulation of a biomarker from one *z*-score to another, rather than an instantaneous switch from a normal to an abnormal level. A key advantage of this formulation is that it can work with purely cross-sectional data because it requires no information about the timescale of change, but instead uses events as control points of piecewise linear segments with arbitrary duration. The model fitting considers increasing number of subtypes *C*, for which we estimate the proportion of subjects *f* that belong to each subtype, and the order $S_C$ in which biomarkers reach each *z*-score for each subtype *c* = 1 … *C*. We determine the optimal number of subtypes *C* for a particular data set through ten-fold cross-validation (see Cross-validation).

**Mathematical model**. The linear *z*-score model underlying SuStaIn is a continuous generalisation of the original event-based model[7,8] which we describe first.

The event-based model in refs. [7,8] describes disease progression as a series of events, where each event corresponds to a biomarker transitioning from a normal to an abnormal level. The occurrence of an event, $E_i$, for biomarker *i* = 1 … *I*, is informed by the measurements $x_{ij}$ of biomarker *i* in subject *j*, *j* = 1 … *J*. The whole data set $\mathbf{X} = \{x_{ij} \mid i = 1 … I, j = 1 … J\}$ is the set of measurements of each biomarker in each subject. The most likely ordering of the events is the sequence **S** that maximises the data likelihood

$$P(\mathbf{X}|\mathbf{S}) = \prod_{j=1}^{J}\left[\sum_{k=0}^{I}\left(P(k)\prod_{i=1}^{k}P\left(x_{ij}|E_i\right)\prod_{i=k+1}^{I}P\left(x_{ij}|\neg E_i\right)\right)\right], \quad (1)$$

where $P(x \mid E_i)$ and $P(x \mid \neg E_i)$ are the likelihoods of measurement *x* given that biomarker *i* has or has not become abnormal, respectively. $P(k)$ is the prior likelihood of being at stage *k*, at which the events $E_1, …, E_k$ have occurred, and the events $E_{k+1}, …, E_I$ have yet to occur. The model uses a uniform prior on the stage, so that $P(k) = 1/(I + 1)$, $k = 0 … I$, i.e. a priori individuals are equally likely to belong to any stage along the progression pattern. The likelihoods $P(x \mid E_i)$ and $P(x \mid \neg E_i)$ are modelled as normal distributions.

The linear *z*-score model we use in this work reformulates the event-based model in (1) by replacing the instantaneous normal to abnormal events with events that represent the (more biologically plausible) linear accumulation of a biomarker from one *z*-score to another. The linear *z*-score model consists of a set of *N* *z*-score events $E_{iz}$, which correspond to the linear increase of biomarker *i* = 1 … *I* to a *z*-score $z_{ir} = z_{i1} … z_{iR_i}$, i.e. each biomarker is associated with its own set of *z*-scores, and so $N = \sum R_i$. Each biomarker also has an associated maximum *z*-score, $z_{max}$, which it accumulates to at the end of stage *N*. We consider a continuous time axis, *t*, which we choose to go from *t* = 0 to *t* = 1 for simplicity (the scaling is arbitrary). At each disease stage *k*, which goes from $t = \frac{k}{N+1}$ to $t = \frac{k+1}{N+1}$, a *z*-score event $E_{iz}$ occurs. The biomarkers evolve as time *t* progresses according to a piecewise linear

function $g_i(t)$, where

$$g(t) = \begin{cases} \frac{z_1}{t_{E_{z_1}}}t, & 0 < t \le t_{E_{z_1}} \\ z_1 + \frac{z_2-z_1}{t_{E_{z_2}}-t_{E_{z_1}}}\left(t-t_{E_{z_1}}\right), & t_{E_{z_1}} < t \le t_{E_{z_2}} \\ \vdots & \\ z_{R-1} + \frac{z_R-z_{R-1}}{t_{E_{z_R}}-t_{E_{z_{R-1}}}}\left(t-t_{E_{z_{R-1}}}\right), & t_{E_{z_{R-1}}} < t \le t_{E_{z_R}} \\ z_R + \frac{z_{max}-z_R}{1-t_{E_{z_R}}}\left(t-t_{E_{z_R}}\right), & t_{E_{z_R}} < t \le 1 \end{cases}.$$

Thus, the times $t_{E_{iz}}$ are determined by the position of the *z*-score event $E_{iz}$ in the sequence **S**, so if event $E_{iz}$ occurs in position *k* in the sequence then $t_{E_{iz}} = \frac{k+1}{N+1}$.

To formulate the model likelihood for the linear *z*-score model we replace Eq. (1) with

$$P(\mathbf{X}|\mathbf{S}) = \prod_{j=1}^{J}\left[\sum_{k=0}^{N}\left(\int_{t=\frac{k}{N+1}}^{t=\frac{k+1}{N+1}}\left(P(t)\prod_{i=1}^{I}P\left(x_{ij}|t\right)\right)\partial t\right)\right], \quad (2)$$

where,

$$P\left(x_{ij}|t\right) = \text{NormPDF}\left(x_{ij}, g_i(t), \sigma_i\right).$$

NormPDF(*x*, *μ*, *σ*) is the normal probability distribution function, with mean *μ* and standard deviation *σ*, evaluated at *x*. We assume the prior on the disease time is uniform, as in the original event-based model.

The SuStaIn model is a mixture of linear *z*-score models, hence we have

$$P(\mathbf{X}|\mathbf{M}) = \sum_{c=1}^{C} f_c P(\mathbf{X}|\mathbf{S}_c),$$

where *C* is the number of clusters (subtypes), *f* is the proportion of subjects assigned to a particular cluster (subtype), and **M** is the overall SuStaIn model.

**Model fitting**. Supplementary Figure 15 provides a flowchart detailing the processes involved in the SuStaIn model fitting. Model fitting requires simultaneously optimising subtype membership, subtype trajectory and the posterior distributions of both. In particular, the cost function here depends on the sequence ordering, which to our knowledge standard algorithms do not handle. We therefore derive our own algorithm to fit SuStaIn, based on the well-established methods developed for the event-based model ([7,8,42,43]), for which we demonstrate convergence and optimality in simulation (see Supplementary Results: Convergence) and in the data sets used here (see Convergence). As shown in the black box in Supplementary Figure 15, the SuStaIn model is fitted hierarchically, with the number of clusters being estimated via model selection criteria obtained from cross-validation. The hierarchical fitting initialises the fitting of each *C*-cluster (subtype) model from the previous *C-1*-cluster model, i.e. the clustering problem is solved sequentially from *C = 1* … $C_{max}$ (where $C_{max}$ is the maximum number of clusters being fitted), initialising each model using the previous model. For the initial cluster (*C* = 1), we use the single-cluster expectation maximisation (E-M) procedure shown in the green box in Supplementary Figure 15, and described subsequently. We fit subsequent cluster numbers (*C* > 1) hierarchically by generating *C-1* candidate *C*-cluster models using the split-cluster E-M procedure shown in the blue box in Supplementary Figure 15, and described subsequently. From these *C-1* candidate *C*-cluster models, the model with the highest likelihood is chosen.

The split-cluster E-M procedure shown in the blue box in Supplementary Figure 15 is used to generate each of the *C-1* candidate *C* cluster models. For each of the *C-1* clusters, the split-cluster E-M procedure first finds the optimal split of cluster *c* into two clusters. To find the optimal split of cluster *c* into two clusters, the data points belonging to cluster *c* are randomly assigned to two separate clusters. The optimal model parameters for these two data subsets are then obtained using the single-cluster E-M procedure (green box in Supplementary Figure 15). These cluster parameters are used to initialise the fitting of a two-cluster model to the subset of the data belonging to cluster *c*, using E-M. This two-cluster solution is then used together with the other *C-2* clusters to initialise the fitting of the *C*-cluster model. The *C*-cluster model is then optimised using E-M, alternating between updating the sequences $\mathbf{S}_c$ for each cluster and the fractions $f_c$. This procedure is repeated for 25 different start points (random cluster assignments) to find the maximum likelihood solution (see Convergence).

The single-cluster E-M procedure shown in the green box in Supplementary Figure 15 is used to find the optimal model parameters (the sequence **S** in which the biomarkers reach each *z*-score) for a single-cluster. In the single-cluster E-M procedure the sequence **S** is initialised randomly. This sequence is then optimised using E-M by going through each *z*-score event *E* in turn and finding its optimal position in the sequence relative to the other *z*-score events, i.e. by fixing the order of the subsequence **T** = **S**/*E* and maximising the likelihood of the sequence by changing the position of event *e* in the subsequence **T**. The sequence **S** is updated until convergence. Again the single-cluster sequence **S** is optimised from 25

different random starting sequences to find the maximum likelihood solution (see Convergence).

**Convergence**. At several points in the model fitting we perform a greedy optimisation from a number of different starting points and choose the maximum likelihood sequence or set of sequences. The multiple runs safeguard against local minima. However, in fact, we find that the optimisation displays good convergence: runs from all start points typically converge to a solution with likelihood within a $1 \times 10^{-3}$ % of the maximum likelihood, and within the uncertainty estimated by the uncertainty estimation procedure (see Uncertainty estimation). The convergence of the SuStaIn algorithm and ability to locate the global minimum and correct solution is further demonstrated in simulation using synthetic data in (Supplementary Results: Convergence).

**Uncertainty estimation**. In addition to estimating the most probable sequence $\mathbf{S}_c$ for each subtype, we can determine the relative likelihood of all sequences for each subtype by evaluating the probability of each possible sequence. This gives us an estimate of the uncertainty in the ordering $\mathbf{S}_c$, which we summarise by plotting the probability that each z-score event appears at each position in the sequence for each subtype. We visualise this probability (see Fig. 2 for example) using different colours to indicate the cumulative probability each region has reached a particular z-score: the cumulative probability of a region going from a z-score of 0-sigma to 1-sigma ranges from 0 in white to 1 in red, the cumulative probability of a region going from a z-score of 1-sigma to 2-sigma ranges from 0 in red to 1 in magenta, and the cumulative probability of a region going from a z-score of 2-sigma to 3-sigma ranges from 0 in magenta to 1 in blue. In practise the number of sequences is too large to evaluate all possible sequences so we use Markov Chain Monte Carlo (MCMC) sampling to provide an approximation to this uncertainty, as in[7,8]. As in refs. [7,8], we take 1,000,000 MCMC samples initialised from the maximum likelihood solution, checking that the MCMC trace shows good mixing properties.

**Cross-validation**. We use ten-fold cross-validation here for two distinct purposes: (i) to evaluate the optimal number of subtypes and (ii) to evaluate the consistency of the subtype progression patterns. We evaluated the optimal number of subtypes using the Cross-Validation Information Criterion (CVIC)[44], i.e. by evaluating the likelihood of each c-subtype model from $c = 1 \ldots C$ on the test data for each fold and choosing the model with the highest out-of-sample likelihood $P(\mathbf{X} \mid \mathbf{M})$, or equivalently the lowest value of the CVIC, across all folds. The CVIC is defined as $\mathrm{CVIC} = -2 \times \log(P(\mathbf{X} \mid \mathbf{M}))$, where $P(\mathbf{X} \mid \mathbf{M})$ is the probability of the data for a particular SuStaIn model, $M$, i.e. $P(\mathbf{X} \mid \mathbf{M}) = \sum_{c=1}^{C} P(\mathbf{X}|\mathbf{S}_c)P(\mathbf{S}_c)$. In cases where the evidence for a more complex model was not strong (a difference of less than 6 between the CVIC and the minimum CVIC across models, or equivalently a difference of less than 3 between the out-of-sample log-likelihood and the minimum out-of-sample log-likelihood across models), we favoured the less complex model to avoid over-fitting[45]. To evaluate the consistency of the subtype progression patterns we performed ten-fold cross-validation by dividing the data into ten folds and re-fitting the model to each subset of the data, with one of the folds retained for testing each time. We report the consistency of the models across folds by computing the similarity between the progression patterns of two subtypes (see Similarity between two subtype progression patterns): the model fitted to each fold and the model fitted to the whole data set.

**Similarity between two subtype progression patterns**. To enable the comparison of subtype progression patterns in data subsets (Supplementary Figure 13B) and across cross-validation folds (CVS in Figs. 2–4, and Supplementary Figures 13A and 14), we measure the similarity of pairs of subtype progression patterns using the Bhattacharyya coefficient[46]. We evaluate the Bhattacharyya coefficient between the position of each biomarker event in the two subtype progression patterns, averaged across biomarker events and MCMC samples. The Bhattacharyya coefficient measures the similarity of the distribution of the position of biomarker events in the subtype sequences and ranges from 0 (maximum dissimilarity) to 1 (maximum similarity).

**Patient subtyping and staging**. We assigned subjects to subtypes and stages predicted by the SuStaIn model (Fig. 5, Tables 1 and 3) by first evaluating the likelihood that they belonged to each subtype (by integrating over disease stage) and choosing the subtype with the highest likelihood, and then evaluating the probability they belonged to each stage of the most probable subtype and choosing the stage with the highest likelihood. When evaluating the likelihood we integrated over the set of MCMC samples to account for the uncertainty in the model parameters, rather than just evaluating the likelihood at the maximum likelihood parameters. This means that a patient's model stage indicates the average position over the posterior distribution on the sequence given the data.

**Strength of assignment to subtype**. We evaluated the strength of an individual's assignment to a particular subtype by comparing the probability that they were at stage ≤2 (i.e. they had no major imaging abnormalities and therefore could not be assigned to a particular subtype) with the probability that they belonged to each SuStaIn subtype (probability for each subtype summed over stages 3+). The strength of the assignment was evaluated as their maximum probability of belonging to one of the subtypes. We considered those with a maximum probability of belonging to a particular subtype of greater than a half as having a strong assignment to a subtype.

**Comparison to subtypes-only and stages-only models**. We compared our SuStaIn model to a subtypes-only model and a stages-only model. In the subtypes-only model, individuals are clustered together into groups based on the similarity of their biomarker measurements—without accounting for heterogeneity in disease stage. The stages-only model is a disease progression model where all subjects are assumed to be samples of a single common progression pattern—without accounting for heterogeneity in disease subtype. We formulated the subtypes-only and stages-only models so that they were as close as possible to the SuStaIn model, but did not model heterogeneity in disease stage or disease subtype, respectively. This allows us to assess the benefit of accounting for this disease stage or subtype heterogeneity in the SuStaIn model. The subtypes-only model consists of a mixture of Gaussians with unknown mean and variance. The subtypes-only model is fitted to symptomatic mutation carriers for GENFI, and AD subjects for ADNI, so that the subtypes correspond to a single diagnostic group. As done for the SuStaIn model, we evaluated the optimal number of clusters (subtypes) using the CVIC[44]. The stages-only model is a special case of the SuStaIn model outlined in Mathematical Model, where only a single subtype is modelled, i.e. $C = 1$.

**SuStaIn modelling of GENFI data set**. Supplementary Tables 4–6 provide a summary of the settings of the SuStaIn algorithm. We applied SuStaIn modelling to various subgroups of the GENFI data set: all 172 mutation carriers, 76 *GRN* mutation carriers, 63 *C9orf72* mutation carriers, 33 *MAPT* mutation carriers. For all mutation carriers we fitted SuStaIn models of up to a maximum of 5 subtypes. For the *GRN, C9orf72* and *MAPT* mutation carriers we fitted SuStaIn models of up to a maximum of 3 subtypes. We chose the z-score events for the GENFI data set to include z-scores of 1, 2 and 3 for each volume, but excluded z-score events where fewer than 10 mutation carriers had values that were greater than that z-score. The maximum z-score, which is reached at the final stage of the progression, was set to be 2, 3 or 5 depending on whether the maximum z-score event was 1, 2 or 3, respectively. We maintained the same z-score events across each of the GENFI experiments.

**SuStaIn modelling of ADNI data set**. We applied SuStaIn modelling to two largely independent (overlap of 59 individuals) subgroups of the ADNI data set: 793 individuals with 3T MRI scans and 576 individuals with 1.5T MRI scans, for which we tested SuStaIn models of up to a maximum of 5 subtypes. As we did for GENFI, we chose the z-score events to include z-scores of 1, 2 or 3 for each volume, but excluded z-score events where fewer than 10 subjects had values that were greater than that z-score. Again the maximum z-score, which is reached at the final stage of the progression, was set to be 2, 3 or 5 depending on whether the maximum z-score event was 1, 2 or 3 respectively. Full details of the settings of the SuStaIn algorithm can be found in Supplementary Tables 4–6.

**Classification of mutation groups using subtypes**. We performed two experiments to compare the ability of subtypes obtained from SuStaIn and the subtypes-only model to classify mutation carriers in GENFI into their different mutation groups. In the first experiment (Tables 1 and 2) we optimised the probability required for assignment to each of the subtypes. This accounts for different amounts of heterogeneity within the different subtypes. In the second experiment (Supplementary Tables 1 and 2) we simply assigned individuals to their most probable subtype and compared their assigned subtype with their mutation group. In both experiments the classification results are reported as out-of-sample accuracies obtained through 10-fold cross-validation.

## Data availability

Genetic FTD data used in this study will become available via the GENFI website (www.genfi.org.uk) and by application to the GENFI data access committee (email: genfi@ucl.ac.uk). The AD data used in this study are available from the Alzheimer's Disease Neuroimaging Initiative (ADNI) database (adni.loni.usc.edu). SuStaIn source code is available at https://github.com/ucl-mig/.

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

## Acknowledgements

A.L.Y. is supported by a Doctoral Prize Fellowship from the EPSRC. N.P.O. is supported by the Biomarkers Across Neurodegenerative Diseases programme, which is funded by The Michael J. Fox Foundation for Parkinson's Research, the Alzheimer's Association, Alzheimer's Research UK and the Weston Brain Institute. R.V.M. is supported by the EPSRC Centre For Doctoral Training in Medical Imaging with grant EP/L016478/1. D.L.T. is supported by the UCL Leonard Wolfson Experimental Neurology Centre (PR/ylr/18575). K.D. is supported by an Alzheimer's Society PhD Studentship. J.B.R. is supported by the Wellcome Trust (103838). G.G.F. was supported by Associazione Italiana Ricerca Alzheimer ONLUS (AIRAlzh Onlus)-COOP Italia. J.D.W. is supported by the Alzheimer's Society, Alzheimer's Research UK and the NIHR UCLH Biomedical Research Centre. S.C. acknowledges the support of the NIHR Queen Square Dementia BRU, ARUK (ART-SRF2010-3), ESRC/NIHR (ES/L001810/1) and EPSRC (EP/M006093/1). J.M.S. acknowl-edges the support of the NIHR Queen Square Dementia BRU, the NIHR UCL/H Biomedical Research Centre, Wolfson Foundation, EPSRC (EP/J020990/1), MRC (MR/L023784/1), ARUK (ARUK-Network 2012-6-ICE; ARUK-PG2017-1946; ARUK-PG2017-1946), Brain Research Trust (UCC14191) and European Union's Horizon 2020 research and innovation programme (Grant 666992). J.D.R. is supported by an MRC Clinician Scientist Fellowship (MR/M008525/1) and has received funding from the NIHR Rare Disease Translational Research Collaboration. The Dementia Research Centre is supported by Alzheimer's Research UK, Brain Research Trust and The Wolfson Foundation. This work is supported by the NIHR Queen Square Dementia Biomedical Research Unit and the NIHR UCL/H Biomedical Research Centre. This work is supported by EPSRC grants EP/J020990/01 and EP/M020533/1 and the *European Union's Horizon 2020 research and innovation programme* under grant agreement No 666992 (EuroPOND: http://www.europond.eu). This work was also supported by the MRC UK GENFI grant (MR/M023664/1). Data collection and sharing for this project was funded by the Alzheimer's Disease Neuroimaging Initiative (ADNI) (National Institutes of Health Grant U01 AG024904) and DOD ADNI (Department of Defense award number W81XWH-12-2-0012). ADNI is funded by the National Institute on Aging, the National Institute of Biomedical Imaging and Bioengineering, and through generous contributions from the following: AbbVie, Alzheimer's Association; Alzheimer's Drug Discovery Foundation; Araclon Biotech; BioClinica, Inc.; Biogen; Bristol-Myers Squibb Company; CereSpir, Inc.;Cogstate; Eisai Inc.; Elan Pharmaceuticals, Inc.; Eli Lilly and Company; EuroImmun; F. Hoffmann-La Roche Ltd and its affiliated company Genentech, Inc.; Fujirebio; GE Healthcare; IXICO Ltd.; Janssen Alzheimer Immunotherapy Research & Development, LLC.; Johnson & Johnson Pharmaceutical Research & Development LLC.; Lumosity; Lundbeck; Merck & Co., Inc.; Meso Scale Diagnostics, LLC.; NeuroRx Research; Neurotrack Technologies; Novartis Pharmaceuticals Corporation; Pfizer Inc.; Piramal Imaging; Servier; Takeda Pharmaceutical Company; and Transition Therapeutics. The Canadian Institutes of Health Research is providing funds to support ADNI clinical sitesin Canada. Private sector contributions are facilitated by the Foundation for the National Institutes of Health (http://www.fnih.org). The grantee organization is the Northern California Institute for Research and Education, and the study is coordinated by the Alzheimer's Therapeutic Research Institute at the University of Southern California. ADNI data are disseminated by the Laboratory for Neuro Imaging at the University of Southern California.

## Author contributions

A.L.Y., D.C.A., J.D.R. and J.M.S. conceived and designed the experiments and wrote the manuscript. A.L.Y. implemented the programming code and analysed the data. N.P.O. and R.V.M. provided feedback on the experiment design. R.V.M. made the brain images in Figs. 1–4, 6 and Supplementary Figures 13–14. M.B. derived the asymmetry measure for GENFI participants. K.Y. advised on sub-scores of the ADAS related to praxic, spatial and memory domains. Members of the ADNI and GENFI consortia recruited patients and collected and pre-processed data. All authors contributed to reviewing and editing of the report.

## Additional information

**Competing interests:** The authors declare no competing interests.

Alexandra L Young [1,2], Razvan V Marinescu [1,2], Neil P Oxtoby [1,2], Martina Bocchetta[3], Keir Yong[3], Nicholas C Firth[1,2], David M Cash [1,3], David L Thomas[4,5], Katrina M Dick[3], Jorge Cardoso [1,3,6], John van Swieten[7], Barbara Borroni[8], Daniela Galimberti[9,10], Mario Masellis[11], Maria Carmela Tartaglia[12], James B Rowe[13], Caroline Graff[14], Fabrizio Tagliavini[15], Giovanni B Frisoni[16], Robert Laforce Jr [17], Elizabeth Finger[18], Alexandre de Mendonça[19], Sandro Sorbi[20,21], Jason D Warren[3], Sebastian Crutch[3], Nick C Fox[3], Sebastien Ourselin[1,3,4,6], Jonathan M Schott [3], Jonathan D Rohrer[3], Daniel C Alexander [1,2], The Genetic FTD Initiative (GENFI) & The Alzheimer's Disease Neuroimaging Initiative (ADNI)

[1]Centre for Medical Image Computing, University College London, London WC1E 6BT, UK. [2]Department of Computer Science, University College London, London WC1E 6BT, UK. [3]Dementia Research Centre, Institute of Neurology, University College London, London WC1N 3BG, UK. [4]Leonard Wolfson Experimental Neurology Centre, UCL Institute of Neurology, University College London, London WC1N 3BG, UK. [5]Neuroradiological Academic Unit, Department of Brain Repair and Rehabilitation, UCL Institute of Neurology, University College London, London WC1N 3BG, UK. [6]School of Biomedical Engineering and Imaging Sciences, King's College London, London WC2R 2LS, UK. [7]Erasmus Medical Center, 3000 CA Rotterdam, The Netherlands. [8]Neurology Unit, Department of Clinical and Experimental Sciences, University of Brescia, 25121 Brescia, Italy. [9]Dept. of Physiopathology and Transplantation, University of Milan, Centro Dino Ferrari, 20122 Milan, Italy. [10]Fondazione IRCCS Ca' Granda Ospedale Maggiore Policlinico, via F. Sforza, 35, 20122 Milan, Italy. [11]Sunnybrook Health Sciences Centre, University of Toronto, ON M4N 3M5, Canada. [12]Centre for Research in Neurodegenerative Diseases, University of Toronto, ON, Toronto M5T 0S8, Canada. [13]University of Cambridge, Department of Clinical Neurosciences, Cambridge CB2 0SZ, UK. [14]Karolinska Institutet, 171 77 Solna, Sweden. [15]Istituto Neurologico Carlo Besta, 20133 Milan, Italy. [16]University Hospitals and University of Geneva, Geneva, Switzerland. [17]Université Laval, Quebec, QC G1V 0A6, Canada. [18]University of Western Ontario, London, ON N6A 3K7, Canada. [19]Faculdade de Medicina, Universidade de Lisboa, 1649-028 Lisboa, Portugal. [20]Department of Neuroscience, Psychology, Drug Research and Child Health, University of Florence, 50121 Florence, Italy. [21]IRCCS Fondazione Don Carlo Gnocchi, Florence, Italy. These authors contributed equally: Jonathan M. Schott, Jonathan D. Rohrer, Daniel C. Alexander.

## The Genetic FTD Initiative (GENFI)

Christin Andersson[22], Silvana Archetti[23], Andrea Arighi[10], Luisa Benussi[24], Giuliano Binetti[24], Sandra Black[25], Maura Cosseddu[26], Marie Fallström[27], Carlos Ferreira[28], Chiara Fenoglio[9], Morris Freedman[29], Giorgio G Fumagalli[9,10,19], Stefano Gazzina[30], Roberta Ghidoni[24], Marina Grisoli[31], Vesna Jelic[32], Lize Jiskoot[33], Ron Keren[34], Gemma Lombardi[19], Carolina Maruta[35], Lieke Meeter[33], Simon Mead[36], Rick van Minkelen[37], Benedetta Nacmias[19], Linn Öijerstedt[38], Alessandro Padovani[39], Jessica Panman[33], Michela Pievani[24], Cristina Polito[40], Enrico Premi[41], Sara Prioni[31], Rosa Rademakers[42], Veronica Redaelli[31], Ekaterina Rogaeva[43], Giacomina Rossi[31], Martin Rossor[3], Elio Scarpini[9,10], David Tang-Wai[34], Hakan Thonberg[44], Pietro Tiraboschi[34] & Ana Verdelho[45]

[22]Department of Clinical Neuroscience, Karolinska Institutet, 171 77 Solna, Sweden. [23]Biotechnology Laboratory, Department of Diagnostics, Civic Hospital of Brescia, 25123 Brescia, Italy. [24]Istituto di Ricovero e Cura a Carattere Scientifico Istituto Centro San Giovanni di Dio Fatebenefratelli,

25125 Brescia, Italy. [25]LC Campbell Cognitive Neurology Research Unit, Sunnybrook Research Institute, Toronto, ON M4N 3M5, Canada. [26]Centre of Brain Aging, University of Brescia, 25121 Brescia, Italy. [27]Department of Geriatric Medicine, Karolinska University Hospital, 171 77 Solna, Sweden. [28]Instituto Ciências Nucleares Aplicadas à Saúde, Universidade de Coimbra, 3000-548 Coimbra, Portugal. [29]Division of Neurology, Baycrest Centre for Geriatric Care, University of Toronto, Toronto, ON M5S 3H7, Canada. [30]Centre of Brain Aging, Neurology Unit, Department of Clinical and Experimental Sciences, University of Brescia, 25121 Brescia, Italy. [31]Fondazione Istituto di Ricovero e Cura a Carattere Scientifico Istituto Neurologico Carlo Besta, 20133 Milan, Italy. [32]Division of Clinical Geriatrics, Karolinska Institutet, 171 77 Solna, Sweden. [33]Department of Neurology, Erasmus Medical Center, 3000 CA Rotterdam, The Netherlands. [34]University Health Network Memory Clinic, Toronto Western Hospital, Toronto, ON M5T 2S8, Canada. [35]Lisbon Faculty of Medicine, Language Research Laboratory, 1649-028 Lisbon, Portugal. [36]MRC Prion Unit, Department of Neurodegenerative Disease, UCL Institute of Neurology, Queen Square, London WC1N 3BG, UK. [37]Department of Clinical Genetics, Erasmus Medical Center, Rotterdam 3000 CA, The Netherlands. [38]Division of Neurogeriatrics, Karolinska Institutet, 171 77 Solna, Sweden. [39]Neurology Unit, Department of Medical and Experimental Sciences, University of Brescia, 25121 Brescia, Italy. [40]Department of Clinical Pathophysiology, University of Florence, 50121 Florence, Italy. [41]Centre for Ageing Brain and Neurodegenerative Disorders, Neurology Unit, University of Brescia, 25121 Brescia, Italy. [42]Department of Neurosciences, Mayo Clinic, Jacksonville, FL 32224, USA. [43]Tanz Centre for Research in Neurodegenerative Diseases, University of Toronto, Toronto, ON M5S 3H7, Canada. [44]Center for Alzheimer Research, Division of Neurogeriatrics, Karolinska Institutet, 171 77 Solna, Sweden. [45]Department of Neurosciences, Santa Maria Hospital, University of Lisbon, 1649-035 Lisbon, Portugal

## The Alzheimer's Disease Neuroimaging Initiative (ADNI)

Michael W Weiner[46], Paul Aisen[47], Ronald Petersen[48], Clifford R Jack[48], William Jagust[49], John Q Trojanowki[50], Arthur W Toga[51], Laurel Beckett[52], Robert C Green[53], Andrew J Saykin[54], John Morris[55], Leslie M Shaw[50], Zaven Khachaturian[47,56], Greg Sorensen[57], Lew Kuller[58], Marc Raichle[55], Steven Paul[59], Peter Davies[60], Howard Fillit[61], Franz Hefti[62], Davie Holtzman[55], M Marcel Mesulam[63], William Potter[64], Peter Snyder[65], Adam Schwartz[66], Tom Montine[67], Ronald G Thomas[47], Michael Donohue[47], Sarah Walter[47], Devon Gessert[47], Tamie Sather[47], Gus Jiminez[47], Danielle Harvey[52], Matthew Bernstein[48], Paul Thompson[68], Norbert Schuff[46,52], Bret Borowski[48], Jeff Gunter[48], Matt Senjem[48], Prashanthi Vemuri[48], David Jones[48], Kejal Kantarci[48], Chad Ward[48], Robert A Koeppe[69], Norm Foster[70], Eric M Reiman[71], Kewei Chen[71], Chet Mathis[58], Susan Landau[49], Nigel J Cairns[55], Erin Householder[55], Lisa Taylor-Reinwald[55], Virginia Lee[50], Magdalena Korecka[50], Michal Figurski[50], Karen Crawford[51], Scott Neu[51], Tatiana M Foroud[54], Steven Potkin[72], Li Shen[54], Kelley Faber[54], Sungeun Kim[54], Kwangsik Nho[54], Leon Thal[47], Neil Buckholtz[73], Marylyn Albert[74], Richard Frank[75], John Hsiao[73], Jeffrey Kaye[76], Joseph Quinn[76], Betty Lind[76], Raina Carter[76], Sara Dolen[76], Lon S Schneider[51], Sonia Pawluczyk[51], Mauricio Beccera[51], Liberty Teodoro[51], Bryan M Spann[51], James Brewer[47], Helen Vanderswag[47], Adam Fleisher[47,71], Judith L Heidebrink[69], Joanne L Lord[69], Sara S Mason[48], Colleen S Albers[48], David Knopman[48], Kris Johnson[48], Rachelle S Doody[77], Javier Villanueva-Meyer[77], Munir Chowdhury[77], Susan Rountree[77], Mimi Dang[77], Yaakov Stern[77], Lawrence S Honig[77], Karen L Bell[77], Beau Ances[55], Maria Carroll[55], Sue Leon[55], Mark A Mintun[55], Stacy Schneider[55], Angela Oliver[55], Daniel Marson[78], Randall Griffith[78], David Clark[78], David Geldmacher[78], John Brockington[78], Erik Roberson[78], Hillel Grossman[79], Effie Mitsis[79], Leyla de Toledo-Morrell[80], Raj C Shah[80], Ranjan Duara[81], Daniel Varon[81], Maria T Greig[81], Peggy Roberts[81], Marilyn Albert[74], Chiadi Onyike[74], Daniel D'Agostino[74], Stephanie Kielb[74], James E Galvin[82], Brittany Cerbone[82], Christina A Michel[82], Henry Rusinek[82], Mony J de Leon[82], Lidia Glodzik[82], Susan De Santi[82], P Murali Doraiswamy[83], Jeffrey R Petrella[83], Terence Z Wong[83], Steven E Arnold[50], Jason H Karlawish[50], David Wolk[50], Charles D Smith[84], Greg Jicha[84], Peter Hardy[84], Partha Sinha[84], Elizabeth Oates[84], Gary Conrad[84], Oscar L Lopez[58], MaryAnn Oakley[58], Donna M Simpson[74], Anton P Porsteinsson[85], Bonnie S Goldstein[85], Kim Martin[85], Kelly M Makino[85], M Saleem Ismail[85], Connie Brand[85], Ruth A Mulnard[72], Gaby Thai[72], Catherine Mc-Adams-Ortiz[72], Kyle Womack[86], Dana Mathews[86], Mary Quiceno[86], Ramon Diaz-Arrastia[86], Richard King[86], Myron Weiner[86], Kristen Martin-Cook[86], Michael DeVous[86], Allan I Levey[87], James J Lah[87], Janet S Cellar[87], Jeffrey M Burns[88], Heather S Anderson[88], Russell H Swerdlow[88], Liana Apostolova[68], Kathleen Tingus[68], Ellen Woo[68], Daniel Hs Silverman[68], Po H Lu[68], George Bartzokis[68], Neill R Graff-Radford[89], Francine Parfitt[89], Tracy Kendall[89], Heather Johnson[89], Martin R Farlow[54], Ann Marie Hake[54], Brandy R Matthews[54],

Scott Herring[54], Cynthia Hunt[54], Christopher H van Dyck[90], Richard E Carson[90], Martha G MacAvoy[90], Howard Chertkow[91], Howard Bergman[91], Chris Hosein[91], Bojana Stefanovic[11], Curtis Caldwell[11], Ging-Yuek Robin Hsiung[92], Howard Feldman[92], Benita Mudge[92], Michele Assaly[92], Andrew Kertesz[93,94,95], John Rogers[93,95], Charles Bernick[93], Donna Munic[93], Diana Kerwin[63], Marek-Marsel Mesulam[63], Kristine Lipowski[63], Chuang-Kuo Wu[63], Nancy Johnson[63], Carl Sadowsky[96], Walter Martinez[96], Teresa Villena[96], Raymond Scott Turner[97], Kathleen Johnson[97], Brigid Reynolds[97], Reisa A Sperling[53], Keith A Johnson[53], Gad Marshall[53], Meghan Frey[53], Barton Lane[53], Allyson Rosen[53], Jared Tinklenberg[53], Marwan N Sabbagh[98], Christine M Belden[98], Sandra A Jacobson[98], Sherye A Sirrel[98], Neil Kowall[99], Ronald Killiany[99], Andrew E Budson[99], Alexander Norbash[99], Patricia Lynn Johnson[99], Joanne Allard[100], Alan Lerner[101], Paula Ogrocki[101], Leon Hudson[101], Evan Fletcher[52], Owen Carmichael[52], John Olichney[52], Charles DeCarli[52], Smita Kittur[102], Michael Borrie[103], T-Y Lee[103], Rob Bartha[103], Sterling Johnson[104], Sanjay Asthana[104], Cynthia M Carlsson[104], Steven G Potkin[69], Adrian Preda[69], Dana Nguyen[69], Pierre Tariot[71], Stephanie Reeder[71], Vernice Bates[105], Horacio Capote[105], Michelle Rainka[105], Douglas W Scharre[106], Maria Kataki[106], Anahita Adeli[106], Earl A Zimmerman[107], Dzintra Celmins[107], Alice D Brown[107], Godfrey D Pearlson[108], Karen Blank[108], Karen Anderson[108], Robert B Santulli[109], Tamar J Kitzmiller[109], Eben S Schwartz[109], Kaycee M Sink[110], Jeff D Williamson[110], Pradeep Garg[110], Franklin Watkins[110], Brian R Ott[111], Henry Querfurth[111], Geoffrey Tremont[111], Stephen Salloway[112], Paul Malloy[112], Stephen Correia[112], Howard J Rosen[46], Bruce L Miller[46], Jacobo Mintzer[113], Kenneth Spicer[113], David Bachman[113], Stephen Pasternak[95], Irina Rachinsky[95], Dick Drost[95], Nunzio Pomara[114], Raymundo Hernando[114], Antero Sarrael[114], Susan K Schultz[115], Laura L Boles Ponto[115], Hyungsub Shim[115], Karen Elizabeth Smith[115], Norman Relkin[59], Gloria Chaing[59], Lisa Raudin[56,59], Amanda Smith[116], Kristin Fargher[116], Balebail Ashok Raj[116], Thomas Neylan[46], Jordan Grafman[63], Melissa Davis[47], Rosemary Morrison[47], Jacqueline Hayes[46], Shannon Finley[46], Karl Friedl[117], Debra Fleischman[80], Konstantinos Arfanakis[80], Olga James[83], Dino Massoglia[113], J Jay Fruehling[104], Sandra Harding[104], Elaine R Peskind[67], Eric C Petrie[106], Gail Li[106], Jerome A Yesavage[118], Joy L Taylor[118] & Ansgar J Furst[118]

[46]UC San Francisco, San Francisco, CA 94143, USA. [47]UC San Diego, San Diego, CA 92093, USA. [48]Mayo Clinic, Rochester, NY 14603, USA. [49]UC Berkeley, Berkeley, CA 94720, USA. [50]UPenn, Philadelphia, PA 9104, USA. [51]USC, Los Angeles, CA 90089, USA. [52]UC Davis, Davis, CA 95616, USA. [53]Brigham and Women's Hospital/Harvard Medical School, Boston, MA 02115, USA. [54]Indiana University, Bloomington, IN 47405, USA. [55]Washington University in St Louis, St Louis, MI 63130, USA. [56]Prevent Alzheimer's Disease 2020, Rockville, MD 20850, USA. [57]Siemens, Munich 80333, Germany. [58]University of Pittsburgh, Pittsburgh, PA 15260, USA. [59]Cornell University, Weill Cornell Medical College, New York City, NY 10065, USA. [60]Albert Einstein College of Medicine of Yeshiva University, Bronx, NY 10461, USA. [61]AD Drug Discovery Foundation, New York City, NY 10019, USA. [62]Acumen Pharmaceuticals, Livermore, CA 94551, USA. [63]Northwestern University, Evanston and Chicago, IL 60208, USA. [64]National Institute of Mental Health, Rockville, MD 20852, USA. [65]Brown University, Providence, RI 02912, USA. [66]Eli Lilly, Indianapolis, IN 46225, USA. [67]University of Washington, Seattle, WA 98195, USA. [68]UCLA, Los Angeles, CA 90095, USA. [69]University of Michigan, Ann Arbor, MI 48109, USA. [70]University of Utah, Salt Lake City, UT 84112, USA. [71]Banner Alzheimer's Institute, Phoenix, AZ 85006, USA. [72]UC Irvine, Irvine, CA 92697, USA. [73]National Institute on Aging, Bethesda, MD 20892, USA. [74]Johns Hopkins University, Baltimore, MD 21218, USA. [75]Richard Frank Consulting, Washington, DC 20001, USA. [76]Oregon Health and Science University, Portland, OR 97239, USA. [77]Baylor College of Medicine, Houston, TX 77030, USA. [78]University of Alabama, Birmingham, AL 35233, USA. [79]Mount Sinai School of Medicine, New York City, NY 10029, USA. [80]Rush University Medical Center, Chicago, IL 60612, USA. [81]Wien Center, Miami, FL 33140, USA. [82]New York University, New York City, NY 10003, USA. [83]Duke University Medical Center, Durham, NC 27710, USA. [84]University of Kentucky, Lexington, KY 0506, USA. [85]University of Rochester Medical Center, Rochester, NY 14642, USA. [86]University of Texas Southwestern Medical School, Dallas, TX 75390, USA. [87]Emory University, Atlanta, GA 30322, USA. [88]University of Kansas, Medical Center, Kansas City, KS 66103, USA. [89]Mayo Clinic, Jacksonville, FL 32224, USA. [90]Yale University School of Medicine, New Haven, CT 06510, USA. [91]McGill University/Montreal-Jewish General Hospital, Montreal, QC H3T 1E2, Canada. [92]University of British Columbia Clinic for AD & Related Disorders, Vancouver, BC V6T 1Z3, Canada. [93]Cognitive Neurology, St Joseph's Health Care, London, ON N6A 4V2, Canada. [94]Cleveland Clinic Lou Ruvo Center for Brain Health, Las Vegas, NV 89106, USA. [95]St Joseph's Health Care, London, ON N6A 4V2, Canada. [96]Premiere Research Institute, Palm Beach Neurology, Miami, FL 33407, USA. [97]Georgetown University Medical Center, Washington, DC 20007, USA. [98]Banner Sun Health Research Institute, Sun City, AZ 85351, USA. [99]Boston University, Boston, MA 02215, USA. [100]Howard University, Washington, DC 20059, USA. [101]Case Western Reserve University, Cleveland, OH 20002, USA. [102]Neurological Care of CNY, Liverpool, NY 13088, USA. [103]Parkwood Hospital, London, ON N6C 0A7, Canada. [104]University of Wisconsin, Madison, WI 53706, USA. [105]Dent Neurologic Institute, Amherst, NY 14226, USA. [106]Ohio State University, Columbus, OH 43210, USA. [107]Albany Medical College, Albany, NY 12208, USA. [108]Hartford Hospital, Olin Neuropsychiatry Research Center, Hartford, CT 06114, USA. [109]Dartmouth-Hitchcock Medical Center, Lebanon, NH 03766, USA. [110]Wake Forest University Health Sciences, Winston-Salem, NC 27157, USA. [111]Rhode Island Hospital, Providence, RI 02903, USA. [112]Butler Hospital, Providence, RI 02906, USA. [113]Medical University South Carolina, Charleston, SC 29425,

USA. [114]Nathan Kline Institute, Orangeburg, NY 10962, USA. [115]University of Iowa College of Medicine, Iowa City, IA 52242, USA. [116]University of South Florida: USF Health Byrd Alzheimer's Institute, Tampa, FL 33613, USA. [117]Department of Defense, Arlington, VA 22350, USA. [118]Stanford University, Stanford, CA 94305, USA

