## [Peer Review File · Nature Communications]

Reviewers' comments:

Reviewer #1 (Remarks to the Author):

This manuscript by Young et al proposes a new algorithm for jointly clustering and staging of patients suffering from a neurodegenerative condition. The underlying assumption is that amongst a set of imaging biomarkers, there is a sequence of events at which each biomarker becomes abnormal. The idea of inferring such a sequence for each cluster, along with detecting the clusters themselves, is a very nice if ambitious idea. The authors propose a likelihood function whose maximization is attempted using a set of heuristics within an expectation maximization (EM) framework.

The strength of the paper is that the problem is cast in a sophisticated statistical way, and inference involves an equally sophisticated, if somewhat arbitrary, sampling approach. The key results suggest there are 4 subtypes in FTD and 3 in AD, each with a well defined temporal progression. These are likely to be very important contributions to the field, if true. I also really liked the principled statistical thinking behind the approach and the ambition of solving such a challenging problem.

However there are many weaknesses that reduce enthusiasm, as listed below.

1. The underlying model of temporal evolution is not well motivated. Why should a bunch of imaging biomarkers follow a set of discrete "stages"? I am not saying this is wrong, just needs to be justified more convincingly. How are the event thresholds specified, and on what basis? If the goal is to develop a joint subtype and stage model, the current one "seems" (admittedly with no supporting evidence) to be needlessly high dimensional. Can the temporal trajectory of each subtype not be parametrized by a low dimensional curve, eg polynomial?

2. Biomarkers are treated as independent of each other, which is at variance with mounting evidence in the field. Different brain regions are connected to each other and pathological progression is known to have a stereotyped and well ordered pattern of spatiotemporal spread. By treating each brain region's atrophy as basically independent, are you losing valuable information that could otherwise have been quite powerful in the inference process?

3. The priors used are pretty uninformative (uniform distributions), and this is at variance to what we already know in the field. Biomarkers are not independent of each other, and perhaps a joint prior or covariance can be specified. At the very least, a Bayesian or Aikike information criterion can and should be added to the model likelihood in order to ensure that the smallest feasible set of parameters is fitted.

4. The optimization procedure (heuristic), while sophisticated and thorough, is quite arbitrary ad hoc and greedy. From my reading the overall algorithm would seem to have no optimality or convergence guarantees, and it is unclear whether there is a global minimum

or a small set of local minima that can be reached by it. There are several combinatorial algorithms with some global optimality properties that could be adapted to solve the present cost function, instead of a mishmash of random initializations, greedy EM and MCMC. Again, I am not doubting that the presented approach works, simply that it is not clear how well it works. From a practical standpoint, proposing a sophisticated likelihood function to be maximized is only half of the job; developing a principled and well-justified algorithm to solve it is equally important. At no point did I get the sense that the authors have thoughtfully considered their algorithmic options and chosen the most suitable ones.

1. Methods:

1. Why are only MR imaging-derived biomarkers specific to atrophy considered in the model? This is quite surprising, since ADNI and I assume GENFI data have multiple PET scans (FDG, PiB, AV45, etc), each of which is a critical biomarker.
2. Although I am mindful that the authors wish to minimize the dimensionality of the problem, it seems quite limiting that only 6 gross regions are considered (frontal, parietal, temporal, occipital, insult and cingulate). What about other important regions like the striatum, hippocampus, entorhinal, cerebellum, etc? Other studies in the field typically present far more regional data (86 in Desikan atlas, 116 in AAL, etc).
3. The method section needs more specificity. Perhaps a table will be useful that contains a list of all relevant quantities and symbols mentioned in the text, their operating range and their initial estimates. List all the biomarkers individually. It is frustrating that the values of R_i , I , N , z_{\max} , C_{\max} , etc are not readily accessible in the manuscript.
4. I think a flowchart of the full algorithm will also aid understanding and readability, as there are multiple heuristics and MCMC sampling steps.

2. Results:

1. Although comprehensive data and statistical analyses were presented, it was still difficult to have full confidence in the key findings (4 clusters in GENFI, 3 in AD). I also appreciate the attempt to provide context by comparing the results to a stages-only and a subtype-only model. Partly, this is the nature of the problem, where a gold standard is simply not available. However, several approaches could have been taken to help place the results in context and improve confidence that what we are seeing is real. Some thoughts are below.
2. A simulation exercise should be undertaken, where artificial biomarker trajectories could be constructed using the likelihood function proposed, and then add relevant noise and other uncertainties. It would be convincing if the inference algorithm could correctly estimate all clusters and stages on this artificial data. In such an approach error rates, MSE, AUC etc can be readily calculated.
3. Similar to #3 above, a plot of performance with respect to increasing dimensionality (e.g. cluster number, number of temporal events N , etc) should be given, and an objective criterion of model evidence (e.g. Bayesian or Aikike information, MDL, etc) provided. This will help the reader understand, for instance, what is the level of evidence supporting the existence of exactly 4 clusters in GENFI and exactly 3 in AD. What happens when you change other hard-wired parameters (N , I , C_{\max})?
4. Identifiability: this term was used in too loose a sense, in my opinion, based simply on Fig 5. There are numerical measures of identifiability that should be reported, if indeed the authors wish to claim this point.

In summary, the present work is potentially a very important contribution to the field of neuroimaging and neurodegeneration, but several aspects require better justification and a more thorough exercising of the results. In view of these points, I would consider its impact to be promising but unvalidated.

Reviewer #3 (Remarks to the Author):

The authors apply machine learning to MRI datasets obtained from patients diagnosed with FTD and AD. They appear to have generated a model that will distinguish the various genetic forms of FTD from each other, and the various subtypes of AD from each other. They also claim to identify two subtypes within one genetic form of FTD relating to the rate of progression. Overall, while I do find that there are some potentially useful pieces of information in this manuscript, it is rather densely presented meaning that it may not have appeal to the general reader and may be more appropriate in a more specialized journal.

There are aspects of this manuscript that do not reflect the typical clinical scenario. For example, they use MRI to identify subjects carrying specific mutations, but most patients are genetically screened for these mutations at the time of presentation, so it is not clear how useful it would be to apply this machine learning approach to identify genetic mutations carriers when either this information is already known or can be much more easily obtained by doing a simple genetic test. Thus, the clinical utility of this approach is unclear.

The authors may be overstating the novelty of their data. It has long been known that GRN mutations produce an asymmetrical pattern of atrophy, that MAPT mutations affect temporal lobes (usually not as asymmetrically), and that C9orf72 related findings are typically widespread and affect cortex and subcortical structures.

It is sometimes difficult to follow exactly what the authors did. For example, it is not clear how many samples were analyzed by their algorithm to generate their model. Did they include normal, pre-symptomatic, or was it just confined to symptomatic cases? These numbers should be included in the first paragraph of the Results section. However, even if all cases were included, then the number of symptomatic cases is still small (e.g. 14 patients carrying GRN mutations, 24 patients carrying C9, 11 MAPT, 117 AD cases). This is likely too small to generate robust results.

With respect to the 117 AD cases, was the diagnosis established pathologically for all of them? It is known that the clinical diagnosis of AD is inaccurate in 15% of cases, so if the diagnosis of AD was based on clinical presentation, did you allow for this inaccuracy in your model?

The programming code used to generate the model as well as the final model should be made available in the supplemental material so that others can test it and add to it as more data becomes available.

We thank the reviewers for their detailed feedback on the manuscript. Addressing their concerns has substantially improved the readability of the manuscript for a general audience, and substantiated the validity of the results. In response to Reviewer 1's concerns over validity of the methodology, we now include a detailed simulation study investigating convergence properties and the effect of the quantities suggested by the reviewer. We have further revised the methods section substantially to improve clarity, with the aid of the flow charts and tables suggested by the reviewer. In response to Reviewer 3 we have revised the manuscript to improve readability for a general audience, clarified the validation role of GENFI, and more generally adapted the text to ensure the novelty and clinical relevance is clear.

Reviewers' comments:

Reviewer #1 (Remarks to the Author):

This manuscript by Young et al proposes a new algorithm for jointly clustering and staging of patients suffering from a neurodegenerative condition. The underlying assumption is that amongst a set of imaging biomarkers, there is a sequence of events at which each biomarker becomes abnormal. The idea of inferring such a sequence for each cluster, along with detecting the clusters themselves, is a very nice if ambitious idea. The authors propose a likelihood function whose maximization is attempted using a set of heuristics within an expectation maximization (EM) framework.

The strength of the paper is that the problem is cast in a sophisticated statistical way, and inference involves an equally sophisticated, if somewhat arbitrary, sampling approach. The key results suggest there are 4 subtypes in FTD and 3 in AD, each with a well defined temporal progression. These are likely to be very important contributions to the field, if true. I also really liked the principled statistical thinking behind the approach and the ambition of solving such a challenging problem.

However there are many weaknesses that reduce enthusiasm, as listed below.

1. The underlying model of temporal evolution is not well motivated. Why should a bunch of imaging biomarkers follow a set of discrete "stages"? I am not saying this is wrong, just needs to be justified more convincingly. How are the event thresholds specified, and on what basis? If the goal is to develop a joint subtype and stage model, the current one "seems" (admittedly with no supporting evidence) to be needlessly high dimensional. Can the temporal trajectory of each subtype not be parametrized by a low dimensional curve, eg polynomial?

R1.1. In fact, SuStaln models imaging biomarkers as following piecewise linear trajectories; a very similar idea to the polynomial trajectories the reviewer suggests. Moreover, SuStaln requires no explicit “event thresholds”, but infers the stage ordering probabilistically. The dimensionality is low - just a few linear segments per biomarker (up to a maximum of four in this work) – and comparable to that of the suggested polynomial model. However, the unique advantage of dividing the trajectories up into stages is that it enables modeling of purely cross-sectional data where the timescale of change is hard if not impossible to estimate. This makes the resulting model practical in clinical scenarios by enabling it to assign subtype and stage to individuals at their first visit. We have rewritten relevant parts of the Methods (see Methods: Mathematical modeling: lines 627-630) and Discussion (see Discussion: Model assumptions and limitations: lines 502-507) to clarify these points.

2. Biomarkers are treated as independent of each other, which is at variance with mounting evidence in the field. Different brain regions are connected to each other and pathological progression is known to have a stereotyped and well ordered pattern of spatiotemporal spread. By treating each brain region’s atrophy as basically independent, are you losing valuable information that could otherwise have been quite powerful in the inference process?

R1.2. First, to clarify, the biomarkers are not independent under the SuStaln model because they co-evolve over time. However, a related assumption, which perhaps the reviewer refers to, is that the biomarker *variance* is assumed independent. This is an assumption common to almost all current disease progression models and is necessary to keep the dimensionality of the problem under control.

We do agree that real world data likely violates this assumption and have now commented on this in the manuscript – see Discussion: Model Assumptions lines 494-497. To demonstrate the validity of the modelling in the presence of biomarker covariance we have performed a simulation study – see Supplementary Material – assessing the effect of different levels of biomarker covariance on the accuracy of the subtype progression patterns recovered by SuStaln (see Figure S7). The effect is subtle and does not affect gross findings such as the presence and broad characteristics of subtypes.

3. The priors used are pretty uninformative (uniform distributions), and this is at variance to what we already know in the field. Biomarkers are not independent of each other, and perhaps a joint prior or covariance can be specified. At the very least, a Bayesian or Aikike information criterion can and should be added to the model likelihood in order to ensure that the smallest feasible set of parameters is fitted.

R.1.3. This is a misunderstanding. The uniform prior is on disease stage and does not relate to biomarker independence. This prior is reasonable given that

the subject-stage is one of the parameters we estimate – the uniform prior adds the least information possible and avoids circularity. With regards to using Bayesian or Akaike information criterion for model selection, we chose to use cross-validation, which does the same job, but is better at penalising complexity – see the Gelman reference in the manuscript. We have adapted the Methods to clarify the role of the prior – see lines 650-652 – and that cross-validation performs the role of model selection – see lines 688-690 and 755-757.

The reviewer's suggestion that one might use model selection (AIC, BIC, or cross validation) to select which biomarker covariances a model might need to include is an interesting idea, but beyond the scope of the current work. We now refer to the possibility in the Discussion (lines 498-500).

4. The optimization procedure (heuristic), while sophisticated and thorough, is quite arbitrary ad hoc and greedy. From my reading the overall algorithm would seem to have no optimality or convergence guarantees, and it is unclear whether there is a global minimum or a small set of local minima that can be reached by it. There are several combinatorial algorithms with some global optimality properties that could be adapted to solve the present cost function, instead of a mishmash of random initializations, greedy EM and MCMC. Again, I am not doubting that the presented approach works, simply that it is not clear how well it works. From a practical standpoint, proposing a sophisticated likelihood function to be maximized is only half of the job; developing a principled and well-justified algorithm to solve it is equally important. At no point did I get the sense that the authors have thoughtfully considered their algorithmic options and chosen the most suitable ones.

R1.4. The optimisation procedure extends a well-established procedure, proposed in Fonteijn et al. *Neuroimage* 2012, further developed in Young et al. *Brain* 2014, and applied in a range of subsequent publications on the event-based model (Oxtoby et al. *Brain* 2018, Wijeratne et al. *Annals of Clinical and Translational Neurology* 2018, Eshaghi et al. *Brain* 2018) to the case of multiple distinct progression patterns. In light of the reviewer's comment we have now revised the manuscript to include a simulation study demonstrating convergence and optimality of the algorithm – see Supplementary Material S1.2.1 – initially excluded for succinctness and appeal to a broad readership. The simulations show very strong convergence and optimality properties, i.e. the EM reliably converges to the global minimum. The purpose of the MCMC is to sample the posterior distribution on the model parameters once the EM finds the maximum likelihood solution; the simulations further demonstrate that estimates of features of the posterior distribution behave sensibly.

With respect, we do not agree that standard combinatorial algorithms adapt naturally to the optimization task we face here, which simultaneously optimizes subtype membership, subtype trajectory, and the posterior

distributions of both. In particular, the cost function here depends on the ordering of a sequence of control points in multiple piecewise linear trajectories (the “events”), which standard algorithms do not handle. We are very happy to be corrected on this point if the reviewer is aware of a specific algorithm for such a task. However, even if such an algorithm does exist, its usage would not affect the findings we report, only reduce the computation time required to obtain them. We add these thoughts to Methods: Mathematical modelling: Model fitting, see lines 681-687.

1. Methods:

1. Why are only MR imaging-derived biomarkers specific to atrophy considered in the model? This is quite surprising, since ADNI and I assume GENFI data have multiple PET scans (FDG, PiB, AV45, etc), each of which is a critical biomarker.

R1.5. We chose to focus on MRI biomarkers alone for a number of reasons:

1. Foremost, sample size. The inclusion of any of the suggested additional biomarkers would reduce the number of available subjects to less than half.
2. Simplicity. MRI alone proves sufficient to demonstrate the capability and potential of subtyping with SuStaln and reveals new knowledge in its own right. Addition of more data types could very well reveal additional and/or refined subtypes, but complicates the message of a paper that already contains substantial novelty. Thus, we prefer to leave such exploration for future work.
3. Finally, clinical utility. Using a single modality has great advantage in the clinical scenario by enabling subtype assignment using (cheap and harmless) MRI alone rather than a complex combination of modalities requiring the use of radiotracers and expensive PET scanners.

We have added these arguments to the manuscript (see Discussion: Disease subtyping and staging: Utility of SuStaln subtypes and stages – lines 482-484), as well as discussion of the potential for the inclusion of additional markers in future work (see Discussion: Disease subtyping and staging: Utility of SuStaln subtypes and stages – lines 484-486).

2. Although I am mindful that the authors wish to minimize the dimensionality of the problem, it seems quite limiting that only 6 gross regions are considered (frontal, parietal, temporal, occipital, insult and cingulate). What about other important regions like the striatum, hippocampus, entorhinal, cerebellum, etc? Other studies in the field typically present far more regional data (86 in Desikan atlas, 116 in AAL, etc).

R1.6. In part there is a misunderstanding here – subcortical regions are included in addition to the 6 larger regions mentioned by the reviewer. As the reviewer points out, increasing biomarkers numbers increases the dimensionality, which complicates model estimation. Since the relatively small number of regions proves sufficient to identify and distinguish subtypes, we see no reason to complicate the analysis with more regions/biomarkers in this initial study. In future, we do plan to include a larger number of regions, as well as additional markers, having first demonstrated validity of the technique in a lower dimensional scenario. We now make this point in lines 484-486.

3. The method section needs more specificity. Perhaps a table will be useful that contains a list of all relevant quantities and symbols mentioned in the text, their operating range and their initial estimates. List all the biomarkers individually. It is frustrating that the values of R_i , I , N , z_{\max} , C_{\max} , etc are not readily accessible in the manuscript.

R1.7. We have added two tables detailing the quantities suggested by the reviewer – see Supplementary Material – Tables S3 and S4 and Methods – Experiments – Lines 826-827.

4. I think a flowchart of the full algorithm will also aid understanding and readability, as there are multiple heuristics and MCMC sampling steps.

R1.8. Thank you for this suggestion. We have added a flowchart of the full algorithm to the manuscript – see Supplementary Material – Figure S15 and updated the Methods section to make the description of the algorithm clearer – see Methods – Model fitting – Lines 680-724.

2. Results:

1. Although comprehensive data and statistical analyses were presented, it was still difficult to have full confidence in the key findings (4 clusters in GENFI, 3 in AD). I also appreciate the attempt to provide context by comparing the results to a stages-only and a subtype-only model. Partly, this is the nature of the problem, where a gold standard is simply not available. However, several approaches could have been taken to help place the results in context and improve confidence that what we are seeing is real. Some thoughts are below.

R1.9. To be clear, we provide other forms of validation apart from the comparison to a subtypes-only and stages-only model that substantially increase confidence in the results: cross-validation of the subtype progression patterns (both datasets), comparison of subtypes estimated without knowledge of FTD genotype to their actual FTD genotype (GENFI), reproducibility of the results in subsets of the data (GENFI), validation in an independent dataset (ADNI), predictive utility of both subtypes and stages

(ADNI). All are strong forms of model validation.

Nevertheless, in response to the reviewer's comments below, the revised manuscript now includes a detailed simulation study to provide further validation of the ability of SuStaln to estimate subtypes and stages – see Results: Synthetic data and Supplementary Material: Simulations.

2. A simulation exercise should be undertaken, where artificial biomarker trajectories could be constructed using the likelihood function proposed, and then add relevant noise and other uncertainties. It would be convincing if the inference algorithm could correctly estimate all clusters and stages on this artificial data. In such an approach error rates, MSE, AUC etc can be readily calculated.

R1.10. As mentioned above, we have performed the simulation study suggested by the reviewer – see Results: Synthetic data and Supplementary Material: Simulations. The results demonstrate and quantify the ability of the SuStaln algorithm to recover subtypes and stages under a variety of different conditions: varying subject numbers, numbers of biomarkers, numbers of subtypes, as well as scenarios that violate the assumptions of the SuStaln model: biomarker covariance, and the presence of a wide spread of progression patterns. The results provide strong additional evidence of the validity of the subtypes we present on the patient data sets.

3. Similar to #3 above, a plot of performance with respect to increasing dimensionality (e.g. cluster number, number of temporal events N , etc) should be given, and an objective criterion of model evidence (e.g. Bayesian or Aikike information, MDL, etc) provided. This will help the reader understand, for instance, what is the level of evidence supporting the existence of exactly 4 clusters in GENFI and exactly 3 in AD. What happens when you change other hard-wired parameters (N , I , C_{\max})?

R1.11. These simulations are now included in the manuscript – see Results: Synthetic data and Supplementary Material: Simulations. We demonstrate that the performance – including the ability to estimate the correct number of clusters – is robust under varying numbers of clusters, biomarkers, and temporal trajectory.

4. Identifiability: this term was used in too loose a sense, in my opinion, based simply on Fig 5. There are numerical measures of identifiability that should be reported, if indeed the authors wish to claim this point.

R1.12. We take the point here that this word has a strong meaning in Bayesian modeling, which was not what we intended here. We have thus changed our terminology throughout the manuscript to talk instead about

assignability of subjects to subtypes (see lines 147 in Introduction, 254-270 in Results and 345, 427, 457-466, 472 in Discussion). Further, we make the concept more quantitative by defining a metric of strength of assignment of an individual to subtype (see Methods: Mathematical Modelling: Strength of assignment to subtype lines 796-804).

In summary, the present work is potentially a very important contribution to the field of neuroimaging and neurodegeneration, but several aspects require better justification and a more thorough exercising of the results. In view of these points, I would consider its impact to be promising but unvalidated.

R1.13. Thank you for the thorough reading and feedback. Addressing these comments has added substantially to confirming the validity and importance of the results and the clarity of the manuscript.

Reviewer #3 (Remarks to the Author):

The authors apply machine learning to MRI datasets obtained from patients diagnosed with FTD and AD. They appear to have generated a model that will distinguish the various genetic forms of FTD from each other, and the various subtypes of AD from each other. They also claim to identify two subtypes within one genetic form of FTD relating to the rate of progression. Overall, while I do find that there are some potentially useful pieces of information in this manuscript, it is rather densely presented meaning that it may not have appeal to the general reader and may be more appropriate in a more specialized journal.

R3.1. We believe the notion of identifying disease subtypes and demonstrating their utility for precision medicine has very wide appeal and application. Moreover, the potential of big-data analytics in healthcare is high in the general scientific consciousness at present; this work represents a unique and substantial advance in our capability to derive new disease understanding from such approaches. It has very wide potential application – to the full spectrum of chronic diseases and beyond – so will be of interest to a broad cross-section of the scientific community. Nevertheless, we agree there were a few dense paragraphs in the main text and have endeavored throughout to move unnecessary technical detail to the Methods or Supplement to enhance accessibility.

There are aspects of this manuscript that do not reflect the typical clinical scenario. For example, they use MRI to identify subjects carrying specific mutations, but most patients are genetically screened for these mutations at the time of presentation, so it is not clear how useful it would be to apply this machine learning approach to identify genetic mutations carriers when either this information is already known or can be much more easily obtained by

doing a simple genetic test. Thus, the clinical utility of this approach is unclear.

R3.2. We believe this comment arises from a misunderstanding of the key message of the manuscript and the role of the FTD results in supporting it. The primary aim is to introduce a new method, SuStaln, for identifying and characterizing disease subtypes from large cross-sectional data sets and to demonstrate its potential in timely applications.

The GENFI dataset experiments, which the comment above refers to, are not intended to demonstrate clinical utility, but rather to provide a validation of SuStaln's ability to uncover known subtypes and their progression patterns. In fact, some new clinical utility does arise from the GENFI models both in temporal staging (not possible with genetics) and as the analysis reveals (for the first time) within-genotype subtypes. However, the main intended clinical utility arising directly from the presented results is in Alzheimer's disease, where SuStaln offers new capability for assigning individuals to subtypes and stages in vivo, with numerous applications, e.g. for selection of cohorts in clinical trials or for precision treatment assignment.

Although we tried to make the roles of these different results clear in the original manuscript, the message was perhaps clouded by the fact that novel findings also emerge from the GENFI analysis. We have now revised several key sentences in the manuscript to clarify that the experiments demonstrating the ability of MRI to distinguish genotype are intended as a validation – see lines 40-42, 137-145, 172-175, 213-216, 274-277, 329-333, 342-343, 469-471 – while the ADNI results demonstrate clinical utility – see lines 145-152, 249-252, 311-323, 345-348, 469-471.

The authors may be overstating the novelty of their data. It has long been known that GRN mutations produce an asymmetrical pattern of atrophy, that MAPT mutations affect temporal lobes (usually not as asymmetrically), and that C9orf72 related findings are typically widespread and affect cortex and subcortical structures.

R3.3. We did not intend to claim these facts as novel findings of our study. The novel findings in FTD are the details of the temporal evolution of these patterns, as well as the presence of distinct within-genotype progression patterns. However, the key novel findings are in AD not FTD. We have revised several sentences in the manuscript to make this clear (in the Summary – see lines 40-42, Introduction – see lines 137-145 and Results – see lines 172-175, 213-216 and Discussion – see lines 329-333, 353-356, 358-362).

It is sometimes difficult to follow exactly what the authors did. For example, it is not clear how many samples were analyzed by their algorithm to generate their model. Did they include normal, pre-symptomatic, or was it just confined to symptomatic cases? These numbers should be included in the first paragraph of the Results section. However, even if all cases were included,

then the number of symptomatic cases is still small (e.g. 14 patients carrying GRN mutations, 24 patients carrying C9, 11 MAPT, 117 AD cases). This is likely too small to generate robust results.

R3.4. We have clarified the subject numbers in the manuscript (see Methods: Experiments: lines 830-832 and 844-845 and Supplementary Material: Table S3) and included them in the first paragraph of the results section as the reviewer suggests (see Results: lines 176,179-180,180-181,183-184).

To clarify the algorithm takes into account all samples in each dataset and is able to leverage information from those in prodromal disease stages, meaning that the number of subjects supporting the GENFI and ADNI results is much greater than the reviewer thought. The GENFI results are derived from the data from all 172 carriers; and the ADNI results are derived from all 793 individuals in the 3T dataset (524 with MCI or AD, in which you would expect to see imaging changes, and a further 269 CN, some of whom may also have prodromal imaging changes), and validated using 576 individuals in the 1.5T dataset (396 with MCI or AD, and 180 CN).

As discussed in R1.9, we have now included extensive simulation results – see Supplementary Material: Simulations – that verify the ability of SuStaln to estimate meaningful subtype progression patterns for datasets of the size of GENFI and ADNI. Of course, it would be preferable to have a larger dataset in the case of FTD. However, in addition to the simulation results, several other validation steps strongly support the findings: cross-validation, reproducibility in subsets of the dataset, ability to classify genotype, and general agreement with previous results as pointed out by the reviewer in their previous point.

With respect to the 117 AD cases, was the diagnosis established pathologically for all of them? It is known that the clinical diagnosis of AD is inaccurate in 15% of cases, so if the diagnosis of AD was based on clinical presentation, did you allow for this inaccuracy in your model?

R3.5. The ADNI AD subjects are not pathologically confirmed. We now include as part of the added simulation study experiments that verify the ability of the algorithm to determine the subtype progression patterns in the presence of small to moderate proportions of misdiagnosed subjects (see Supplementary Material: Simulations: Results: Misdiagnosis and Figures S9 and S10).

The programming code used to generate the model as well as the final model should be made available in the supplemental material so that others can test it and add to it as more data becomes available.

R3.6. We will make the code available once the manuscript is published. This is standard practice in our group; see <https://github.com/ucl-mig>.

REVIEWERS' COMMENTS:

Reviewer #1 (Remarks to the Author):

This revision by Young et al has substantially addressed many although not all reviewer comments. Explanatory text has been added, and is for the most part appropriate and targeted to the critiques. Most importantly, a large new simulation study is now being reported in SI, which demonstrates desirable properties of the algorithm, like convergence, robustness to errors, covariance amongst biomarkers, etc. These simulations are especially useful because the proposed algorithms are largely heuristic, with no optimality guarantees. The revision has also fully addressed one of my key issues, of model selection and the use of criteria such as Bayesian or information criteria. They have instead used cross validation to achieve the same purpose, and this is appropriate.

I believe in view of these substantial changes and clarifications, and the sophistication of the original approach, this revision merits consideration for possible publication.

One additional suggestion I would add, if the editors find it appropriate, is that the authors add a clear table of exactly which biomarkers are used - a full list of all demographic, tissue, cognitive and regional atrophy biomarkers should be listed. This critical information is very poorly presented in the current version, especially of the regional biomarkers.

Reviewer #3 (Remarks to the Author):

I have no further comments.

REVIEWERS' COMMENTS:

Reviewer #1 (Remarks to the Author):

This revision by Young et al has substantially addressed many although not all reviewer comments. Explanatory text has been added, and is for the most part appropriate and targeted to the critiques. Most importantly, a large new simulation study is now being reported in SI, which demonstrates desirable properties of the algorithm, like convergence, robustness to errors, covariance amongst biomarkers, etc. These simulations are especially useful because the proposed algorithms are largely heuristic, with no optimality guarantees. The revision has also fully addressed one of my key issues, of model selection and the use of criteria such as Bayesian or information criteria. They have instead used cross validation to achieve the same purpose, and this is appropriate.

I believe in view of these substantial changes and clarifications, and the sophistication of the original approach, this revision merits consideration for possible publication.

One additional suggestion I would add, if the editors find it appropriate, is that the authors add a clear table of exactly which biomarkers are used - a full list of all demographic, tissue, cognitive and regional atrophy biomarkers should be listed. This critical information is very poorly presented in the current version, especially of the regional biomarkers.

Thank you for your constructive feedback on the manuscript. We have added the additional table of biomarkers suggested – see Supplementary Table 4.

Reviewer #3 (Remarks to the Author):

I have no further comments.